# Non-human primates can flexibly learn serial sequences and reorder context-dependent object sequences

**Xuan Wen[1,2], Adam Neumann[1], Seema Dhungana[1], Thilo Womelsdorf** [1,2,3*]

**1** Department of Psychology, Vanderbilt University, Nashville, Tennessee, United States of America,
**2** Vanderbilt Brain Institute, Nashville, Tennessee, United States of America, **3** Department of Biomedical Engineering, Vanderbilt University, Nashville, Tennessee, United States of America

* thilo.womelsdorf@vanderbilt.edu

## Abstract

Intelligent behavior involves mentally arranging learned information in novel ways and is particularly well developed in humans. While nonhuman primates (NHP) will learn to arrange new items in serial order and re-arrange neighboring items within that order, it has remained contentious whether they are capable to re-assign items more flexibly to non-adjacent serial positions. Such mental re-indexing is facilitated by inferring the sequential structure of experiences as opposed to learning serial chains of item-item associations. Here, we tested the ability for flexible mental re-indexing in rhesus macaques. Subjects learned to choose five objects in a pre-determined sequential order. A change of the background context indicated when the object order changed, probing the subjects to mentally re-arrange objects to non-adjacent positions of the learned serial structure. Subjects successfully used the context cue to pro-actively re-index items to new, non-adjacent positions. Mental re-indexing was more likely when the initial order had been learned at a higher level, improved with more experience of the re-indexing rule and correlated with working memory performance in a delayed match-to-sample task. These findings suggest that NHPs inferred the sequential structure of experiences beyond a chaining of item-item associations and mentally re-arrange items within that structure. The pattern of results indicates that NHPs form non-spatial cognitive maps of their experiences, which is a hallmark for flexible mental operations in many serially ordered behaviors including communication, counting or foraging.

## Introduction

Mental flexibility refers to the ability to arrange thoughts or actions in novel ways. Flexible mental operations are required for many serially ordered behaviors including communication, counting, problem solving, and foraging [1,2]. These serial behaviors benefit from knowing the underlying sequential structure that guides the ordering of

**Data availability statement:** The data supporting this study are openly available at the Open Science Framework repository: https://osf.io/2vk97/ (DOI: https://doi.org/10.17605/OSF.IO/2VK97). All analysis code for reproducing the main and supplementary figures is publicly available on GitHub: https://github.com/xwen1765/Re-ordering-of-Sequential-Order-in-Nonhuman-Primates.

**Funding:** This work was supported by the National Institute of Mental Health (R01MH123687 to TW), url: https://www.nimh.nih.gov. The funders had no role in study design, data collection and analysis, the decision to publish, or the preparation of this manuscript.

**Competing interests:** The authors have declared that no competing interests exist.

**Abbreviations :** DMTS, delayed match-to-sample task; FDR, false discovery rate; MCMC, Markov Chain Monte Carlo; M-USE, multi-task suite for experiments; NHP, nonhuman primates.

information. In language systems, communication can rely on the flexible arrangement of words into sentences based on syntactical rule structure; Counting and mathematical operations rely on knowing the regular numbering systems; Solving problems is facilitated by the ordering of subgoals reflecting the problem structure; and foraging becomes efficient if a spatial map structure of patch locations is retained to ease choosing the right foraging patch. For all these examples, mental operations and behavior become more flexible when the underlying structure for sequencing information is known. Consistent with this suggestion, the ability to infer sequential structures from experiences and to mentally re-arrange information within that structure underlies measures of intelligence and is particularly well developed in humans compared to nonhuman primates (NHP) [3,4]. However, it has remained debated, how capable NHPs are to infer abstract sequential structure and whether they can use that structure to flexibly guide mental operations [5–8].

Powerful experimental paradigms for testing how subjects infer sequential structure from their experiences involve the serial ordering of visual objects [9,10]. Studies that use serial order learning paradigms have shown that NHPs are able to order objects in sequence and mentally swap the order or nearby (adjacent) objects in a sequence. In particular, NHPs understand the relative ordinal position of objects in multi-item sequences [11–18], are able to swap object positions in a sequence when they are adjacent to each other [19,20], and recall items of a 3-item long sequence backwards in reverse order [21,22]. These abilities highlight that NHPs show some flexibility of mentally manipulating rank-ordered items, but these experimental results also point to limitations of NHPs, when compared to humans, to extract sequential structure among visual items beyond a rank-ordering of items, suggesting their mental abilities are limited to the serial chaining of nearby items [5,23]. For example, the ability to recall a sequence A-B-C backwards as C-B-A involves the re-ordering of items to new positions in the sequence [21], but this re-ordering can be achieved by swapping the relative rank of adjacent items in a sequence without the need to represent an abstract sequential structure or ordinal positions to which items are flexibly assigned [24].

Here, we set out to investigate whether NHPs are capable to mentally re-order items of 5-item sequences to new, non-adjacent positions within that sequence. NHPs learned sequences of objects A-B-C-D-E and after they learned the initial sequence were probed with a context cue to re-order non-adjacent items B and D to a novel A-D-C-B-E sequence. The re-ordering of objects to different positions in the sequence can be achieved by representing the specific object items as independent information that is assigned to an ordinal position of a sequential structure as opposed to representing a chain of object-object pairs irrespective of an underlying ordinal structure [21]. In computational models such a temporal sequential structure of experienced environments can be described as a non-spatial cognitive map of item locations [7,25]. A non-spatial cognitive map can be inferred from experiences and stored in synaptic weights in long-term memory, or they can be retained in ongoing activity slots using working memory [26]. We hypothesized that rhesus monkeys will be able to infer such a non-spatial structure from experience and that this structure

will be revealed when they proactively re-order non-adjacent object items B and D to a novel A-D-C-B-E sequence. To test whether such an ability is influenced by longer term memory and by working memory, we will test sequential behavior if the NHPs at different levels of experience, and relate their performance to their short-term memory performance as assessed with a different task.

We tested the learning and flexible re-ordering of object sequences in four rhesus macaques using 3-D rendered objects shown on a touchscreen mounted in a Kiosk station in their home cage [27]. The paradigm required NHPs to choose five objects in a predetermined sequential order as shown in S1 Movie (Fig 1A). Objects were shown simultaneously on the screen. Subjects learned the sequential order of the five objects over successive trials. In each trial they received immediate visual feedback (a halo around the chosen object) about whether they chose an object at the correct ordinal position. When the choice was incorrect, they had to re-touch the last correctly chosen object. In each trial they received fluid reward when they completed the five-object sequence. When they did not complete the sequence within 15 trials (maximally 10 errors) they did not receive fluid reward. When a sequence was completed, or the maximum of 15 choices was reached the subjects were presented with a new trial in which the same objects were arranged at new locations to prevent a spatial strategy. During performance, subjects received for each correctly chosen object immediate visual feedback (yellow halo; for incorrect choices: grey halo) and an increment (for incorrect choices: a decrement) of the slider position that signaled how many steps away subjects were from receiving fluid reward (Fig 1A, *see* S1 Movie).

After 15 trials with the same sequence, the background (context 1) changed to a new background (context 2) that showed the same objects but required a new pre-determined sequential order with objects B and D swapping positions. Thus, the onset of the 'context 2' background served as a cue for the subjects to re-order the object sequence they had learned in the preceding 15 trials. The task paradigm allowed assessing the sequence learning (in context 1) and the re-ordering of the learned sequence (in context 2). A 'context 1 + 2' pair used the same set of multidimensional objects [28]. Ten to 12 unique 'context 1 + 2' pairs were evaluated in an individual experimental session that lasted approximately 90 min. Half-way through a session, after four or five context 1 + 2 sequence pairs, the subjects performed for 120 trials a delayed match-to-sample task that assessed short-term memory (Fig 1B). After the delayed match-to-sample task subjects performed an additional five to six context 1 + 2 sequence pairs. Two or three of these late performed context 1 + 2 sequence pairs were the same as shown earlier in the session (i.e., they were repeated sequences), which allowed assessing longer-term memory effects.

We found that subjects gradually learned to complete 5-object sequences A-B-C-D-E in context 1. When the background changed to context 2 to indicate that the same objects are re-ordered to A-D-C-B-E they overall followed two different strategies in parallel. In some trials in context 2 they chose at the second serial position in context 2 the object C instead of B, i.e., they recognized that object B needed to be re-ordered but erroneously chose the next-in-line object C instead of the correct object D. The erroneous choice of C at the second ordinal position suggests subjects erroneously inferred the next adjacent objects in the series, i.e., they followed a serial inference strategy. In other trials in context 2 subjects correctly anticipated the swapped positions of objects B and D and chose object D at the second ordinal position. This pro-active swapping was more likely than a serial inference strategy when the initial sequence was learned better, and it correlated with working memory performance on a session-by-session level. The findings suggest that NHPs effectively use context cues to mentally re-index objects within a learned sequential structure.

## Results

### Learning of object sequences

Four rhesus monkeys performed the sequence learning task in on average 30.25 experimental sessions (subject B: 11; J: 40; K: 59; S: 11). To assess whether subjects successfully acquired the 5-object sequences we calculated the proportion of correct choices at each ordinal position in the last four trials of the 15 trials that subjects performed each sequence in

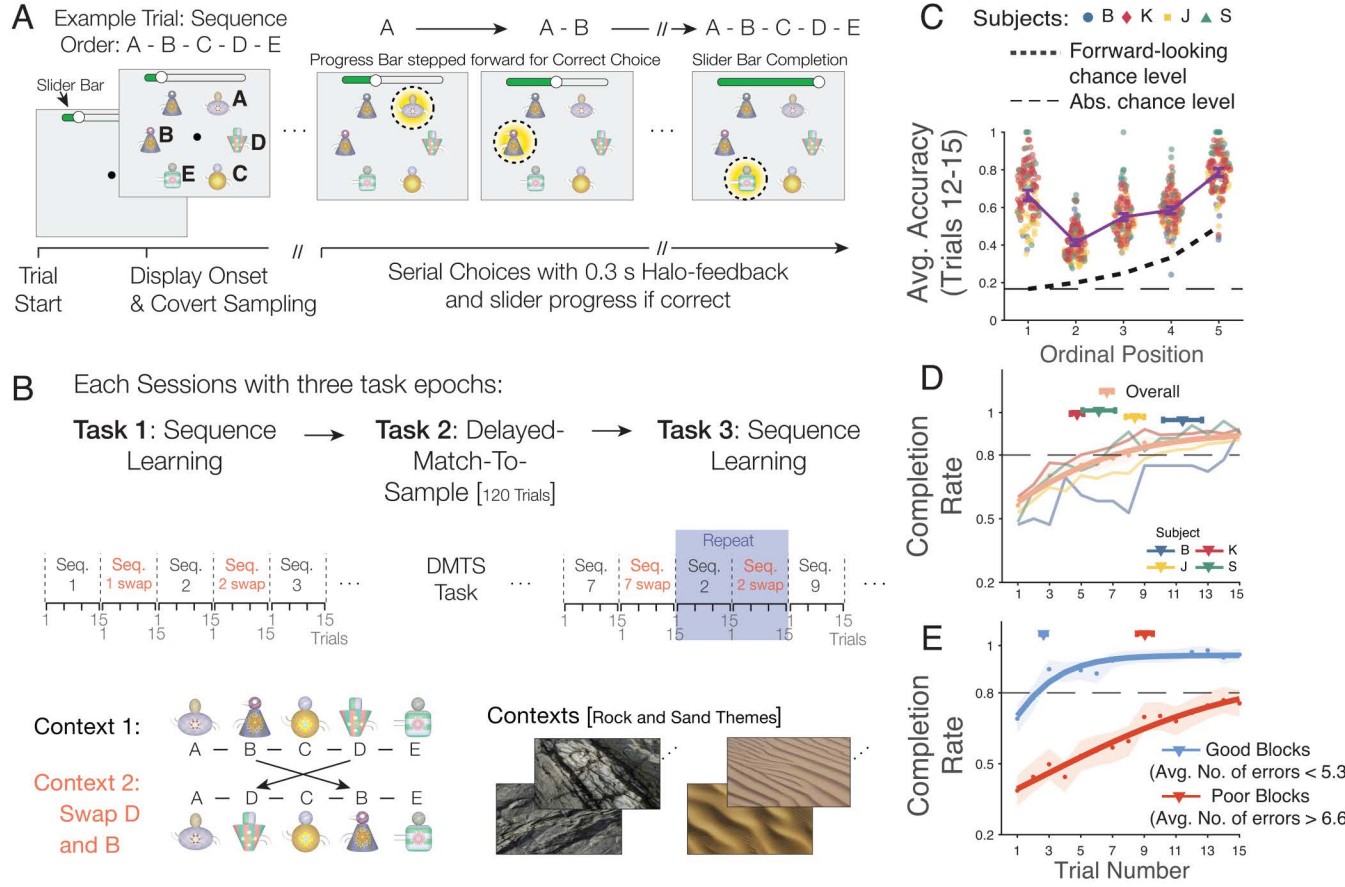

## Figure 1

**Fig 1. Learning 5-object sequences. (A)** Each trial presented six objects. Monkeys learned to touch five objects in a pre-determined order A-B-C-D-E and avoid a distractor object. A correct choice led to visual feedback (yellow halo) and incremented the slider position of a progress bar on top of the screen. Monkeys received fluid reward upon completion of the sequence. **(B)** Each sequence was presented in 15 separate trials on the same 'context 1'. Then the context changed and the object position 2 (object B) and 4 (object D) swapped (*red* font). After 5–6 "context 1 – context 2" pairs subjects performed a delayed match-to-sample (DMTS) task for 120 trials. Then, sequence learning was performed again with approximately 3 new and approximately 3 repeat-sequence pairs from earlier in the session. Sequences were presented on rock- or sand- themed contexts. **(C)** Prop. of correct choices for each ordinal position in context 1 averaged across trials 12–15. Darker dashed line indicates forward-looking chance level, i.e., assuming subjects do not consider reselecting previously correct objects. Lighter dashed line is absolute chance levels (1/6). Error bars are 95% conf. int. **(D)** Avg. rate of completing the 5-object sequences. Symbols mark the first trial (±SE) at which participants completed a sequence on average in 80% of all trials, with this level of performance sustained for at least three consecutive subsequent trials. **(E)** Avg. completion rate for the top third of 'good' sequences with lowest error rate (<5.33 errors) and for the bottom third of 'poorer' performed sequences (>6.6 errors). Red and blue triangles represent the mean trials to reach completion (±95% CI) for good and poor blocks, respectively, considering only blocks that reached completion threshold. The data underlying this figure can be found in the S1 Data file.

context 1. We found that all subjects performed above chance at each ordinal position documenting successful learning (proportion correct at ordinal position 1 (OP 1): 0.67 (mean) ± 0.026 (95% conf.); OP 2: 0.41±0.016; OP 3: 0.55±0.020; OP 4: 0.59±0.017; OP 5: 0.79±0.020) (Figs 1C and S1A). To assess the learning of a sequence we evaluated not individual ordinal positions but evaluated how likely subjects completed a full sequence by calculating how many trials the subjects needed to complete the 5-object sequences on average in 80% of trials above chance levels, which was with

10 or less erroneous choices. Subjects reached this 80% completion rate on average within 6.62 ± 0.37 trials (subject B: 11.40 ± 1.23; J: 8.39 ± 0.58; K: 4.74 ± 0.40; S: 6.11 ± 1.03) (Fig 1D). Sequence learning performance is visually shown in the S1 Movie. These results show that subjects gradually learned the 5-object sequences over multiple trials. There was variability of learning, which allowed rank ordering the sequences by the average number of errors per trial across trials for each sequence and comparing learning for fewest versus most errors. For the top third of sequences with the least errors per trial (≤5.33 errors per trial) the 80% completion rate was achieved after 2.68 (±0.20) trials, while the bottom third of sequences with the highest error rates (>6.6 errors per trial) reached 80% completion only after 9.25 (±0.51) trials (Fig 1E).

Learning gradually progressed through all five ordinal positions, reaching 80% accuracy of the first ordinal position on average after 1.02 trials (±0.03, 95% confidence interval), and of the second to fifth ordinal position on average after trials 1.24 (±0.09), 1.81 (±0.19), 2.60 (±0.33), and 3.69 (±0.45) (S1B and S1C Fig). Learning was achieved by reducing erroneous choices of objects that were not yet chosen, while perseverative errors or violations of the task rule (rule: '*if an error was made, re-chose the last correctly chosen object*') were infrequent (S1D and S1E Fig). Reaction times gradually increased from second to fifth ordinal position, consistent with prior studies (S2 Fig) [29].

**Subjects pro-actively re-order non-adjacent objects of well-learned object sequences**

We next tested whether subjects could re-assign objects of a learned A-B-C-D-E sequence to different ordinal positions. When subjects completed 15 trials of the sequence in *context 1*, we changed the context background to a new *context 2* and re-ordered the same objects to the new order A-D-C-B-E. The sequence in context 2 swapped positions of objects B and D (Fig 2A, for example performance see S1 Movie). Overall, subjects initially dropped in performance in context 2 for a few trials but then adjusted to the swapped sequence in context 2 quickly, completing the A-D-C-B-E sequences in context 2 at 80% completion rate on average at trial 3.43 (±0.56, Mean ± 95% CI), compared to 6.62 (±0.72) trials for the initial learning of the A-B-C-D-E sequences in context 1 (S2A Fig). The faster adjustment to the swapped sequence order in context 2 compared to new learning of a sequence in context 1 was evident in each subject (completion rate for sequences of context 1/2 in subject B: 5.91 (±0.89)/4.27 (±0.80); J: 4.71 (±0.53)/2.67 (±0.31); K: 2.98 (±0.26)/1.97 (±0.17); S 5.91 (±0.94)/3.27 (±0.60) (S2B Fig). Reaction times were similar across ordinal positions for context 1 and 2 (S2C and S2D Fig).

How did the subjects adjust to the swapped positions of object B and D in context 2? A serial chaining framework predicts subjects will erroneously choose object B at the second ordinal position and adjust to this error by choosing the next neighboring object C at the third ordinal position because its relative position is closest to the second position, i.e., they inferred the serially next ordinals' position object. We refer to this behavior as reflecting a serial inference strategy. In contrast to such serial inference, subjects may also at the second ordinal position pro-actively choose object D. This would reveal they understood that D was indexed to the absolute fourth position in context 1 and needed to be re-arranged in context 2 to the new A-D-C-B-E sequence. We found subjects used both strategies, but more likely showed a re-indexing of object D from the fourth ordinal position in context 1 to the second ordinal position in context 2 (Figs 2 and 3). To quantify the use of both strategies (serial inference and re-indexing), we first analyzed the average choices in context 2 during the learning phase, i.e., before subjects reached the 80% completion rate in context 2. When subjects erroneously choose object B in context 2 they were more likely to adjust to this error by correctly choosing next object D rather than object C (Welch's *t* test, $p = 2.9 \times 10^{-9}$) (Fig 2B). We consider this finding to indicate retro-active re-indexing in response to error feedback.

We next ask whether re-indexing of object D to the second ordinal position in context 2 varied with the depth of learning the initial sequence in context 1, hypothesizing that re-indexing is more likely when the initial sequence in context 1 was learned at a higher proficiency. We found that retro-active re-indexing of object D in context 2 was more likely when the initial A-B-C-D-E sequence in context 1 was learned at a higher level, i.e., for the one third of sequences

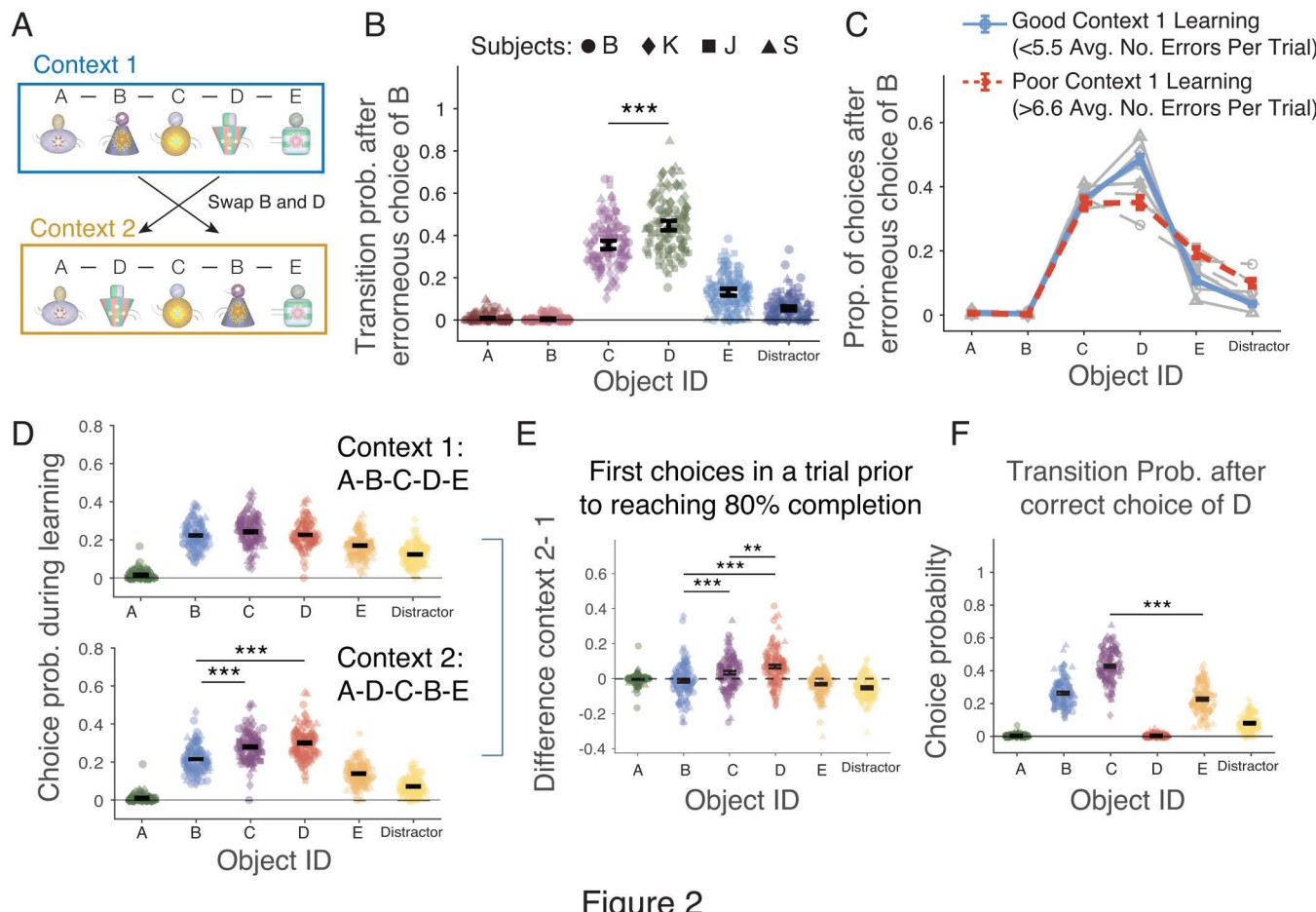

Figure 2

**Fig 2. Subjects swap objects between non-adjacent ordinal positions. (A)** Context 2 swapped the ordinal position of object B and D. **(B)** Probability of choosing objects immediately after erroneously choosing object B in context 2: for A: 0.007 ± 0.003 (95% CI); B: 0.004 ± 0.002; C: 0.356 ± 0.020; D: 0.450 ± 0.022; E: 0.130 ± 0.014; Distractor: 0.054 ± 0.010. Three stars denote $p < 0.001$ difference (Welch's $t$ test with Bonferroni correction). **(C)** Choosing object D after an error on B ('retro-active swapping') was more likely when the context 1 sequence was learned better (*blue:* one third of good performed sequences with low (≤5.33) errors per trial) than when they were learned poorly (*red:* one third of sequences with highest number of (>6.6) errors per trial). **(D)** Choice probability for the first touched object after correctly choosing A in context 1 (*upper*), context 2 (*lower*) in trials prior to reaching the 80% completion rate. **(E)** The difference of the upper and lower panels in D. Subject more likely chose D over C and B in context 2. **(F)** Choice probability for the first touched object after correctly choosing D in context 2. Stars denote significance levels (Welch's $t$ test with Bonferroni correction). Error bars are SEs. The data underlying this figure can be found in the S1 Data file.

with the lowest error rates in context 1 (<5.33 errors per trial) (avg. proportion for choosing objects A: 0.0067 ± 0.0018 (Mean ± SE); B: 0.0041 ± 0.0014; C: 0.3609 ± 0.0109; D: 0.4856 ± 0.0113; E: 0.1078 ± 0.0070; Distractor: 0.0349 ± 0.0042) (Fig 2C). When context 1 was performed poorly, i.e., for the one third of sequences with the highest error rate (>6.6 errors per trial), subjects similarly often applied the correct re-indexing strategy (choosing object D at the second position) and the incorrect serial-inference strategy (choosing object C at the second position) (avg. proportion for choosing object A: 0.0058 ± 0.0029; B: 0.0015 ± 0.0015; C: 0.3480 ± 0.0182; D: 0.3509 ± 0.0182; E: 0.1944 ± 0.0151; Distractor: 0.0994 ± 0.0114) (Fig 2C).

In a second analysis, we quantified whether subjects used the change in contexts (the appearance of context 2) to swap objects pro-actively, i.e., whether – after correctly choosing A in context 2 – they chose object D at the second ordinal position without erroneously choosing object B. We found that compared to context 1, in context 2 subjects more likely

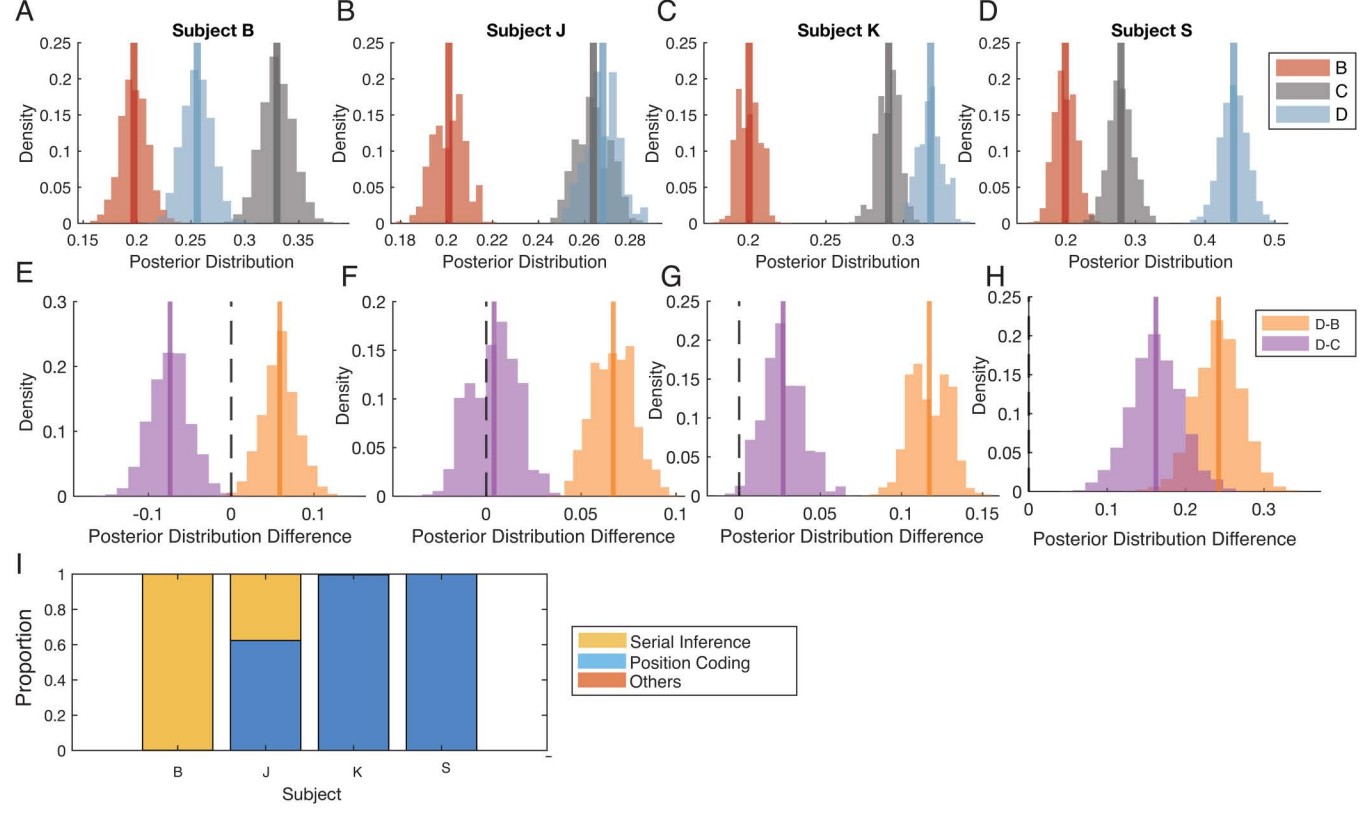

**Figure 3**

**Fig 3. Bayesian analysis quantifies the choice strategies of individual subjects during sequence re-ordering. (A–D)** Posterior estimations of choice probability of choosing object B, C and D after choosing object A in context 2. **(E–H)** Estimated posterior distribution differences of the sequence of choices of object D to B and D to C. All subjects showed above 0 for the choice sequence D→B, two subjects showed above 0 for the choice sequence D→C. **(I)** Proportion of the use of different strategies in context 2 at the second ordinal position across sessions. The serial inference strategy correspond to P(C) > P(D) and P(C) > P(B); the swapping (position coding) strategy corresponds to: P(D) > P(B) and P(D) > P(C); other strategies were not evident in any session but could have included: P(B) > P(C) and P(B) > P(D) and P(C) > P(D). Three subjects (subjects J, K, and S) showed predominantly the swapping strategy (position coding), while one subjects (subject B) used the serial inference strategy (Subject J: position coding: 62.4%, serial inference: 37.6%; Subject K: position coding: 99.6%, serial inference: 0.4%; Subject S: position coding: 100.0%, serial inference: 0.0%; Subject B: position coding: 0.1%, serial inference: 99.9%). Other strategies were not evident in any subject. The data underlying this figure can be found in the S1 Data file.

chose object D than B, indicating pro-active re-indexing of the object D from a non-adjacent fourth position in context 1 to the second position in context 2, but subjects also more likely chose object C than B, indicating the wrong serial inference strategy (D versus B, and C versus B, *t*-tests, both *p*<0.001; Fig 2D and 2E). Directly comparing object D and C choices revealed, however, that after correctly choosing object A, subjects more likely chose object D (indicating re-indexing) than object C (indicating serial inference) (Welch's *t* test, *p*<0.01; Fig 2D and 2E). This pattern of results suggests that subjects considered both, object C and object D as possible target objects in context 2 when adjusting to the new context (prior to reaching 80% completion rate), but that they more likely applied the re-indexing strategy anticipating that object D was the correct object at the second position.

One prediction of the re-indexing strategy is that once object D was correctly chosen in context 2 subjects should next choose object C, instead of erroneously choosing object E which followed object D in context 1. Consistent with this hypothesis subjects chose C more likely than object E following choosing D, i.e., they left object E at the fifth ordinal

position (Fig 2F) (Welch's *t* test, $p < 1 \times 10^{-10}$; correctly choosing object C: $0.415 \pm 0.010$, incorrectly choosing object E: $0.232 \pm 0.009$ and object B: $0.263 \pm 0.008$).

To address whether the proactive swapping strategy is consistent across subjects, we applied a hierarchical Bayesian model to estimate, for each individual, the probability of selecting objects B, C, or D immediately after choosing A in context 2 (Fig 3A–3D). For each of the subjects, the probability of choosing D at the second ordinal position was consistently higher than that of choosing object B, while in three of four subjects object D was chosen more likely than object C consistent with swapping as the dominant strategy (Fig 3E–3H; Subject B: B = $0.20 \pm 0.01$ (Mean ± SE), C = $0.33 \pm 0.01$, D = $0.26 \pm 0.01$; Subject J: B = $0.20 \pm 0.01$, C = $0.26 \pm 0.01$, D = $0.27 \pm 0.01$; Subject K: B = $0.20 \pm 0.01$, C = $0.29 \pm 0.01$, D = $0.32 \pm 0.01$; Subject S: B = $0.20 \pm 0.02$, C = $0.28 \pm 0.02$, D = $0.44 \pm 0.02$). One of the subjects (subject B) chose object C more likely than object D at the second ordinal position suggesting a predominance of the serial inference strategy over the swapping strategy. To evaluate whether the overall result pattern across subjects would distinguish the two choice strategies, we aggregated the results in context 2 across subjects and found that the posterior probability that the choice probability for D exceeded that for B was 0.99, and the posterior probability that choosing object D exceeded choosing C was 0.657. These results show the highest probability was that subjects choose D over both, objects B and C at the second ordinal position, consistent with pro-active swapping of object D into the second ordinal position being the prevalent strategy.

We further examined whether proactive swapping was the dominant retrieval strategy at the level of individual experimental sessions, or whether swapping was mixed with serial inference or other strategies. To address this, we fit three candidate strategies to the choice probabilities for objects B, C, and D in context 2 in each estimation: swapping positions, i.e., position coding (object D more likely being chosen than B and C), serial inference (object C more likely being chosen than B and D) and others (neither position coding nor serial inference) (Fig 3I). The analysis revealed that swapping D and B as the choice strategy was favored in the majority of sessions, with 65.68% of sessions showing a posterior model probability for this strategy. The serial inference strategy accounted for 34.23% of sessions.

The swapping results so far indicate that subjects adjusted their choices in context 2 consistent with re-indexing objects D and B to different ordinal positions, but this re-indexing might depend on the efficient use of the immediate visual feedback subjects received for correct and erroneous choices. For example, when subjects chose object D at the correct second ordinal position in context 2, they received positive feedback, which on average will have increased the likelihood of continuing choosing object D as opposed to erroneously choosing object C at the correct second ordinal position. To control for this possibility and test whether the re-indexing was evident without having received feedback we performed a third analysis restricted to only the first choice of the first trial in context 2 at the second position (i.e., after object A was correctly chosen) and compared it with the first choice of the last trial in context 1 under the same condition (Fig 4). Across all sequences, 'first choices' in the first trial of context 2 were more likely to object D than object B (Welch's *t* test with FDR correction, $p = 0.0227$) as well as more likely to object C than B (Welch's *t* test, $p = 0.0227$) (Fig 4B). Comparison of choices in context 2 (first choice in first trial) with context 1 (first choice in last trial) showed that subjects not only chose C and D more likely than object B, but that they chose object D more likely than object C (Welch's *t* test, $p = 0.0026$) (Fig 4C).These findings show that overall subjects used two different strategies in context 2 already in their first choices after the context change: a serial inference (choosing object C as first choice) and the re-indexing of object D as first choice at the second ordinal position. The direct comparison with context 1 suggests that the re-indexing strategy was more prevalent than the serial inference strategy (Fig 4C). This pattern of result was found when considering all sequences (Fig 4A–4C), and also when considering only the subset of two-third of all sequences that were learned best in context 1 (with on average <6.6 errors per trial in context 1) (Fig 4D–4F). However, neither the pro-active re-indexing strategy, nor the serial inference strategy were evident when considering poorly performed sequences, i.e. when restricting the analysis to the subset of one-third of sequences that were performed poorly in context 1 (with on average >6.6 errors per trial in context 1) (Fig 4G–4I).

   

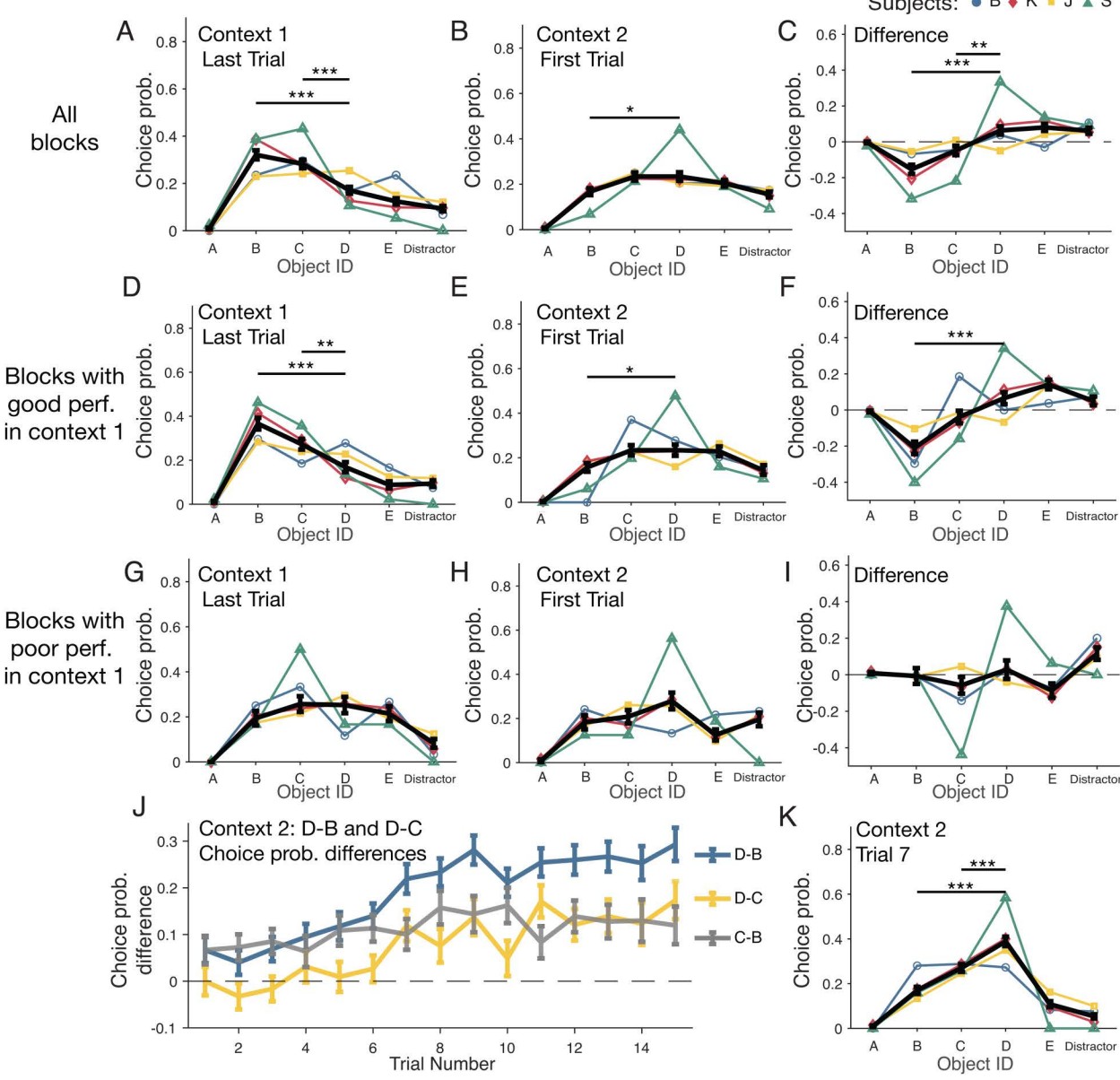

**Fig 4. Subjects pro-actively re-index object order prior to choice feedback when sequences were learned well. (A)** Choice probabilities for the first choice in the last trial of context 1 (after choosing object A). **(B)** Choice probabilities for the first choice of the first trial in context 2 (after choosing object A). Object D was chosen more frequently than B. **(C)** Difference in choice probabilities between the first choice of the first trial of context 2 and the last trial of context 1. Stars denote $p < 0.05$ (*), $p < 0.01$ (**), and $p < 0.001$ (***) (Welch's t test with FDR correction). **(D–F)** Same format as **A–C** for the third of blocks with the best performance in context 1 (<5.33 errors per trial, top two-thirds of blocks). **(G–I)** Same format as A–C for the third of blocks with the poorest performance in context 1 (>6.6 errors per trial, bottom third of blocks). **(J)** Trial-by-trial differences of the 'first choice' choice probabilities after subjects chose correctly object A context 2. The blue line shows the difference of the likelihood to choose D rather than B ('D–B'), while the other lines show the difference of the likelihood to choose C rather than B (grey: 'C–B') and to choose rather than C (yellow: 'D–C'). Object B was less likely chosen than C and D as first choice in the first trial of context 2, and object C was less likely chosen than D starting from the seventh trial. **(K)** Choice probabilities of the seventh trial in context 2 across all blocks. Object D was chosen more frequently than object C. The data underlying this figure can be found in the S1 Data file.

Subject performed the swapped sequence in context 2 for 15 trials, allowing us to track the first choices at the second ordinal position across trials to quantify when choosing object D (re-indexed to the second ordinal position) reliably exceeded choosing object C (erroneous choosing the next adjacent object from the third ordinal position). This analysis confirmed that overall, subjects already avoided object B at the second ordinal position in the first choice in the first trial of context 2, but it took on average seven trials in context 2 for subjects to choose object D more likely that object C (Fig 4J and 4K).

## Sequence memory and rule memory improves swapping ability

To swap the position of objects in a sequence depends on (1) recalling the original sequence from memory and (2) on understanding the swapping rule. We tested how long-term memory affected swapping performance by evaluating performance of sequences in the second half of the session after subjects completed the delayed match-to-sample task. Subjects performed five to six context 1+2 sequence pairs in this second half of the experimental sessions (Fig 1B). Half of these 'late' context 1+2 sequence pairs were novel ('New Late' sequences), but the other half were identical to a context 1+2 sequence pair that was shown prior to the delayed match-to-sample task ('Repeat' sequences). If long-term memory facilitated performance and swapping of positions B and D in context 2 of the late sequences, then we expect that late, repeated sequences were performed better and showed more prominent pro-active swapping than late, new sequences. We found that subjects reached the 80% completion rate already after on average 1.29 (±0.13 95% CI) trials for repeated sequences, which was faster than the on average 3.5 (±0.54) trials subjects needed to learn the same sequence early in the session, and it was also faster than the 2.90 (±0.47) trials needed to reach 80% completion rate for new sequences that were interspersed late in the session ('New Late' sequences) to control for a possible effect of the time-in-task (Figs 5A and S3). The likelihood of pro-active swapping was already above chance for context 2 of the early sequences and did only marginally increase further in repeated sequences, evident as a statistical trend (New Early versus Repeat: $p = 0.0717$; New Early versus New Late: $p = 0.1178$ Repeat versus New Late: $p = 0.7913$; Fig 5B). However, in repeated sequences subjects were significantly more likely in context 2 to retro-actively swap object D into the second position after erroneously choosing object B at the second ordinal position (New Early versus Repeat: $p = 0.0014$; New Early versus New Late: $p = 0.0020$; Repeat versus New Late: $p = 0.3133$; Fig 5C). This pattern of results shows that memory of the sequence from earlier in the session facilitated error correction in trials when subjects failed to pro-actively swap object D into the second ordinal position.

Next, we analyzed whether the depth of memorizing the initial sequence early in the session influenced the swapping performance of that sequence later in the session. We found that subjects more likely pro-actively swapped D into the second position in context 2 for repeated sequences late in the session when the sequences were better memorized early in the session, i.e., when they were performed at a higher completion rate ($R^2$: 0.02, $p = 3 \times 10^{-8}$) (Fig 5D and 5E). Similarly, retro-active swapping of object D after erroneously choosing B in context 2 at the second position was also more likely late in the session when the initial sequence was better learned early in the session ($R^2$: 0.0079, $p = 0.0017$) (Fig 5F and 5G).

These findings show that learning a sequence early in the session at a higher performance level predicted not only improved retro-active swapping in context 2 early in the session as described above (Fig 2C) and late in the session (Fig 5F and 5G), but also improved the ability to pro-actively swap objects B and D in context 2 late within the session (Fig 5D and 5E).

In addition to these findings of within-session memory effects, there is also the possibility that swapping abilities change with experience across sessions. Such an across-session effect would be expected if subjects improve in applying the swapping rule. We found that pro-active swapping was apparent already in the first experimental sessions and gradually increased over sessions ($R^2$: 0.0717, $p = 0.0030$; Fig 5H), while the likelihood of retro-active swapping remained similar across sessions ($R^2$: 0.0108, $p = 0.2559$) (Fig 5I).

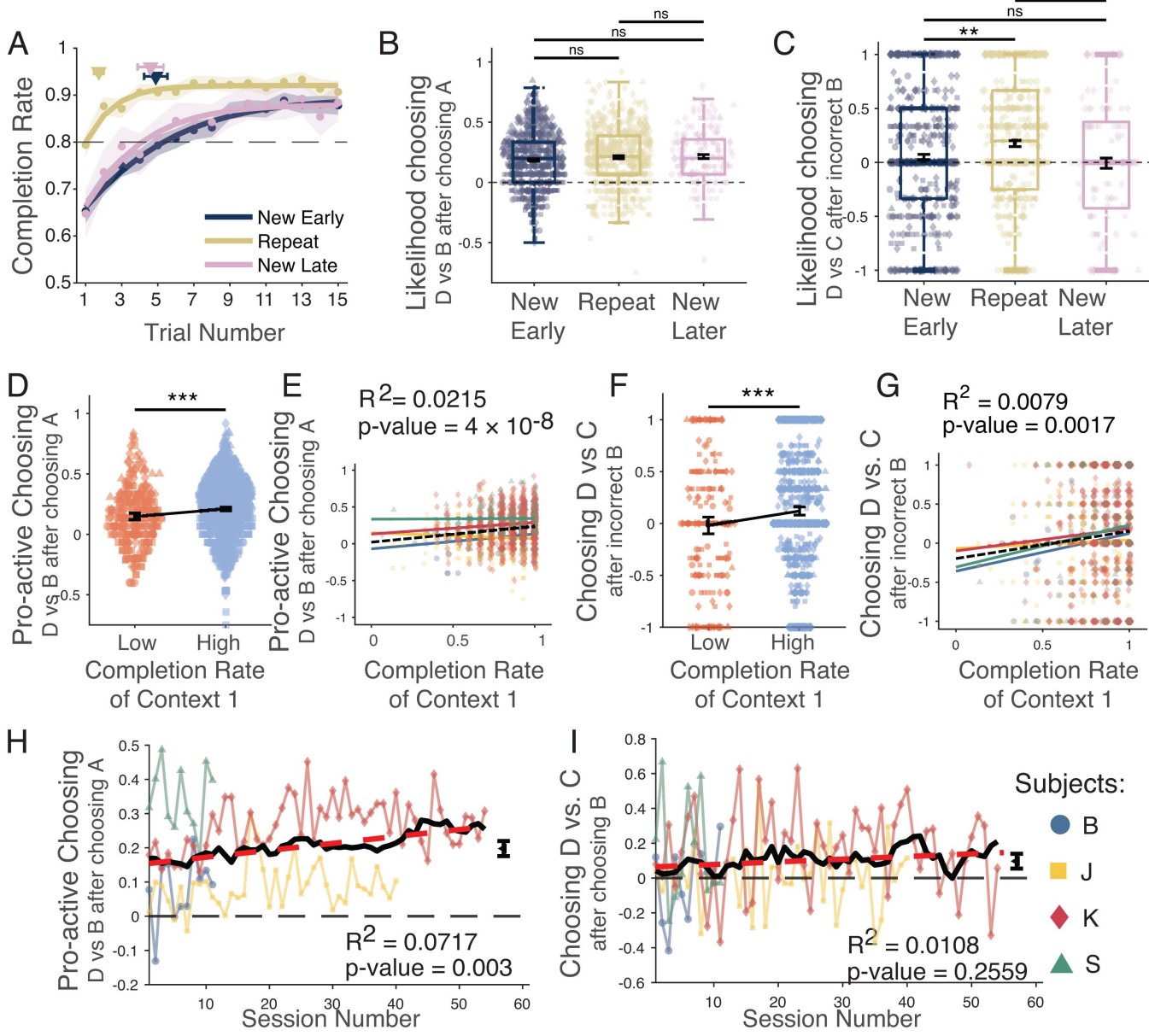

**Fig 5. Memory of object sequences improves pro-active and retro-active swapping performance. (A)** Completion rate of sequences before the working memory task (New Early), and of new and repeated sequences performed after the working memory task (New Late, Repeat). **(B)** Pro-active swapping in context 2 (values above 0 reflect that D is more likely being chosen than B after choosing A): Anticipating that D is in second position in context 2 is similarly evident in all conditions. **(C)** Retro-active swapping after erroneously choosing object B in context 2 (values above 0 reflects that D is more likely being chosen than C after choosing B): Difference of choosing object D vs. C. **(D)** Better performance in context 1 (blue *vs.* red: higher *vs.* lower completion rate) is associated with higher likelihood of pro-actively swapping object D and B in context 2. (Poor performed sequences: M = 0.1471, 95% CI [0.1201, 0.1740]; well performed sequences: M: 0.2102, 95% CI [0.1964, 0.2240]; *p*: 5.2 × 10⁻⁵). **(E)** Regression of the completion rate in context 1 and the likelihood to choose object D immediately after object A in context 2 (*β*: 0.2136; intercept: 0.0215; $R^2$: 0.0214; *p*: 3.67 × 10⁻⁸; Cohen's *f²*: 0.0220). The different colors represent individual subjects. The dashed black line is the regression calculated across all subjects' data points. **(F)** Same format as **D** for retro-active swapping, i.e., for choosing object D in context 2 at the second ordinal position after erroneously choosing B (poorly performed sequences: M: −0.0194, 95% CI [−0.0999, 0.0611]; well performed sequences: M: 0.1202, 95% CI [0.0809, 0.1594]; *t* test, *p*: 0.0024). **(G)** Regression of context 1 completion rate and retro-active swapping in context 2 (*β*: 0.3502; intercept: −0.1968; $R^2$: 0.0079; *p*: 0.0017; Cohen's *f²*: 0.0079). The colors are fits to data from different subjects; the dashed black regression line is based on data from all subjects combined. **(H)** Choosing object D proactively after object A in context 2 was more likely than choosing object B across 124 valid sessions (M: 0.20, 95% CI [0.18, 0.22]; *p* < 1 × 10⁻¹⁰). The effect increased in strength over sessions (regression *β* value: 0.0020; *p*: 0.0030; Cohen's *f²*: 0.0773). **(I)** Choosing object D after an erroneous

choice of B in context 2 was more likely than choosing object C on average across 121 valid sessions (M: 0.10; 95% CI [0.06, 0.14], one-sample $t$ test, $p = 1.8 \times 10^{-5}$). Regression slope (red line) was not significant ($\beta$ value: 0.0015; $p$: 0.2559; Cohen's $f^2$: 0.0110). The data underlying this figure can be found in the S1 Data file.

### Pro-active swapping of object positions correlates with working memory

Computational work has shown that learning a sequence can be similarly achieved with long term memory that retains sequential relations in synaptic weights, or with working memory that retains sequential relations in recurrent network activity [26]. According to these insights the subject's ability to retain objects in working memory might be predictive of the overall speed of learning sequences or the ability to flexibly swap objects in the sequence. Consistent with this later suggestion recent neurophysiological findings support the suggestion that swapping objects from later and earlier ordinal positions involves transiently storing the position index of the original objects in a temporary variable, which is similar to a short-term memory buffer [21]. We thus hypothesized that pro-active swapping performance can be predicted from subject's working memory ability. We tested this hypothesis by assessing working memory with a delayed match-to-sample in the same behavioral sessions that assessed swapping (Figs 6A and S4). We found that working memory performance did not correlate with sequence learning accuracy (Fig 6B), but working memory accuracy significantly correlated with pro-active swapping abilities on a session-by-session basis (Fig 6C). This result indicates that successfully swapping object D and B in context 2 is not only influenced by longer term sequence-memory (Fig 5), but also by an ability to hold objects active in working memory.

We next tested whether memory of the context background in which a sequence was learned influenced swapping behavior. When a sequence was repeated after the working memory task, it was presented on either the same, or on a different contextual background than the initial new sequence. We found improved sequence learning performance on repeated versus early (initial) sequences when the context of the repeated sequence was the same compared to when the repeated sequence was shown on a different context background (Welch's $t$ test, $p = 0.0083$; Fig 6D). However, this overall contextual facilitation did not modulate pro-active or retro-active swapping in context 2 (pro-active sapping: Welch's $t$ test: $p = 0.3556$; Fig 6E; retro-active swapping: Welch's $t$ test: $p = 0.1225$; Fig 6F).

## Discussion

We found that rhesus monkeys learned in experimental sessions 6–8 new 5-object sequences, i.e., 4–5 sequences before the delayed match to sample task and 2–3 new sequence of the 5–6 sequences shown after the delayed match-to-sample task) (Fig 1 and S1 Movie). When a change in context indicated that non-adjacent objects B and D switched positions, three of four subjects swapped these objects more likely than erroneously choosing the next-ranked object C in the sequence (Figs 2B–2F and 3). The swapping occurred pro-actively, i.e., prior to making an error (Figs 4A–4F, 5B, 5D, 5E, and 5H), as well as retro-actively, i.e., when subjects corrected an erroneous choice of B at the second ordinal position in context 2 (Fig 5C, 5G, and 5I). Four factors were associated with better swapping performance: First, retro-active swapping in context 2 was more likely when the sequence had been learned at a higher proficiency level in the immediately preceding context 1 (Figs 2C and 4D–4F). Secondly, pro-active swapping was more likely in repeated sequences in context 2 when context 1 was performed at higher proficiency level approximately 30–60 min earlier within the same experimental session (Fig 5D–5G). Thirdly, swapping became more likely over the course of the experiment when the swapping rule had been performed for more sessions (Fig 5H). Fourth, pro-active swapping was positively correlated with a subject's working memory performance as assessed with a separate delayed match-to-sample task (Fig 6C). Consistent with the importance of successful memorization of the sequence in context 1 for swapping behavior, the one subject (subject B) who showed in context 2 swapping behavior less likely than a serial inference (i.e., continued choosing object C more likely than D after choosing A) was the subject with the poorest learning performance in context 1 by reaching 80% completion rate on average on trial 11.40 compared to 4.74–8.39 trials in the other monkeys (Fig 1D).

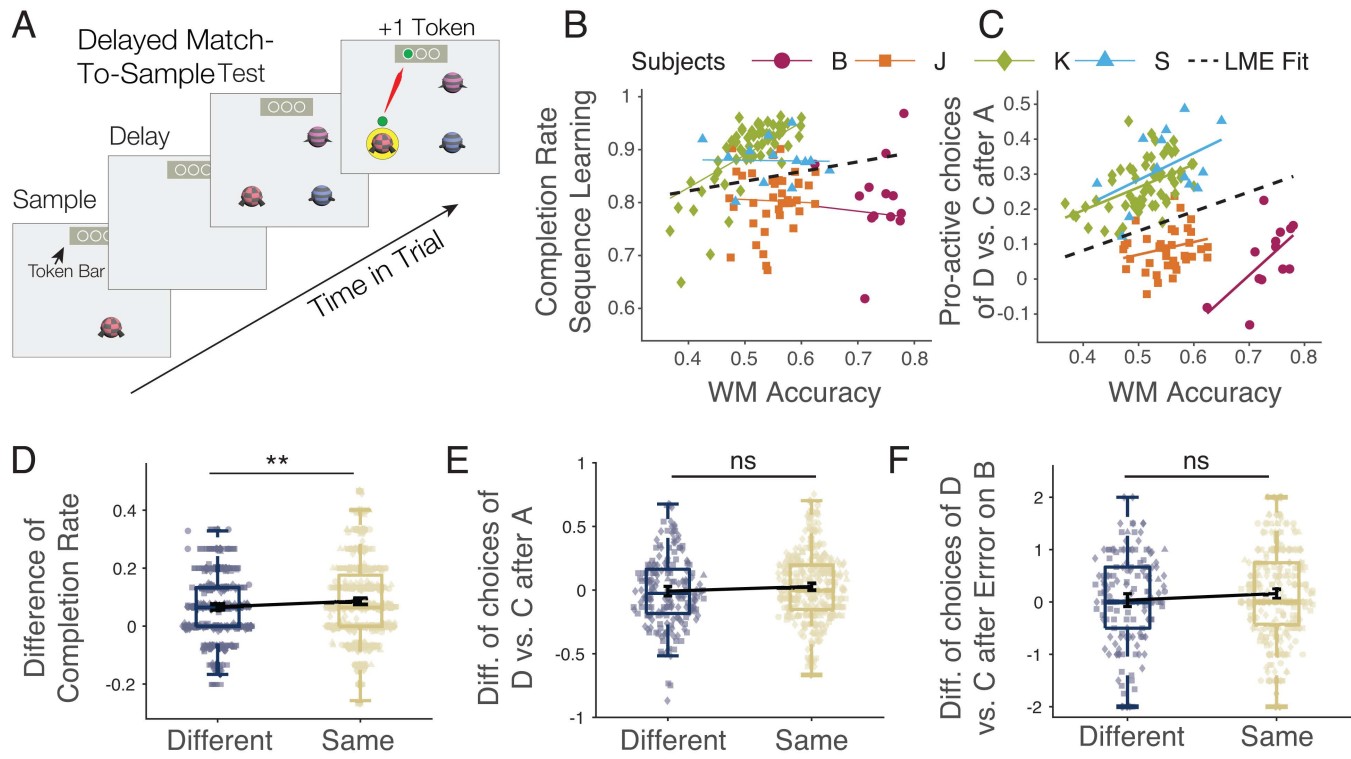

## Figure 6

**Fig 6. Working memory performance correlates with pro-active re-indexing performance. (A)** Delayed match-to-sample (DMTS) paradigm. **(B)** Across sessions, the DMTS accuracy did not correlate with the completion likelihood of sequences. **(C)** DMTS accuracy correlated with pro-active swapping, i.e., with choosing object D after A in context 2. Black line denotes avg. Linear Mixed Effect model; colors show individual subjects. **(D)** Contexts facilitated sequence learning: Completion rates were higher for sequences repeated with the same context ($n = 669$, *yellow*) than a different context ($n = 427$, *blue*): same contexts: Mean ± 95% CI: 0.0868 ± 0.0104; different contexts: 0.0662 ± 0.0111l Welch's *t* test: $p = 0.0083$. **(E, F)** Same and different contexts in repeated sequences resulted in similar pro-active swapping performance (E: same context: Mean ± 95% CI: 0.0374 ± 0.0279; Different context: −0.0083 ± 0.0286; Welch's *t* test: $p = 0.3556$), and similar retro-active swapping (F: same context: Mean ± 95% CI: 0.0374 ± 0.0279; different context: 0.0083 ± 0.0286; Welch's *t* test: $p = 0.1225$). The data underlying this figure can be found in the S1 Data file.

Taken together, these results show that rhesus monkeys infer the temporal order of objects during sequence learning and are able to swap the index of object identities to absolute positions in that latent order when a new context instructed them to reassign objects to new temporal positions.

### Swapping behavior reflects flexible mental re-indexing of object associations

Successful pro-active swapping behavior shows that monkeys used the context of the swapped sequence as a cue to re-order objects A-B-C-D-E of context 1 to a new A-D-C-B-D order in context 2. A neuronal correlate for such cue-triggered reconfiguration process has recently been suggested in an experiment that required NHPs to recall backwards the spatial order of 2- and 3-item sequences [21]. Groups of frontal cortex neurons represented forward spatial 2-item sequences A-B until a cue instructed the subjects to recall the sequence backwards. During the re-coding of the

PLOS Biology

forward into a backward sequence neuronal population activity in frontal cortex transiently encoded both, the forward A-B sequence and the swapped items A as a rank 2 object and B as a rank 1 object. This transient encoding of both, forward and backward item order, was then followed by neuronal populations representing only the backward sequence that needed to be recalled [21]. These findings suggest that swapping the order of adjacent objects involves a short-term memory that contains in separate neuronal populations the old item order (forward sequence) and the new item order (the backward item order). While these neuronal findings are limited to playing backwards adjacent items, we suggest that they provide a versatile framework to conceptualize the re-indexing of objects to non-adjacent positions in our study. In particular, this framework predicts that in our study objects of the A-B-C-D-E sequence in context 1 will be encoded by neurons indexing their ordinal position [22]. When the context changes and objects D and B need to be swapped, the original index of objects B and D are temporarily encoded in a short-term buffer, and the original A-B-C-D-E sequence is reconfigured into the new A-D-C-B-E sequence. When this operation completes, the transient buffer is not needed anymore and the swapped sequence is encoded by a group of neurons that is partially distinct from the group of neurons that encoded the initial sequence [21]. There are two key insights of this framework. First, it suggests that the swapping operation involves the re-indexing of objects to their absolute serial position within a sequence, and second, that swapping involves the temporary short-term memory of old and new object positions in sequence. The indexing operation requires that the objects and the sequential structure are independently encoded in the neuronal network that performs this operation, which is well supported by neurophysiological evidence [21,22]. Our results are consistent with this framework and suggest that NHPs are able to utilize these neuronal re-indexing processes when learning to swap objects to non-adjacent sequential positions. This suggestion goes beyond frameworks, such as the serial chaining framework, that suggest that sequences and sequential re-ordering involves the formation of new serial chains of item-item associations. Taken together, the prevalence of pro-active swapping of non-adjacent object positions in our study documents a high level of flexibility of mental operations, consistent with a re-indexing of objects to an abstract sequential structure. In our study, the sequential structure is a straight line of item positions to which objects are indexed. Such a line constitutes one of the simplest forms of a non-spatial cognitive map, which is implicated to be the cornerstone of flexible cognition [7,26,30].

### Mental re-indexing of object positions is linked to working memory abilities

We found that pro-active swapping in context 2 was more likely in sessions in which subjects also showed better delayed match-to-sample task performance, suggesting a relationship of working memory and pro-active swapping (Fig 6C). In contrast, working memory did not correlate with the average speed of learning the sequence (Fig 6B). This pattern of results is consistent with the suggestion that learning a sequence initially involves associative mechanisms while swapping relies on mentally manipulating a learned structure in working memory. Consistent with this suggestion the re-indexing framework postulates a transient short-term memory representation of the to-be-swapped objects is needed to re-arrange the order of a sequence [21]. More generally, a short-term memory enables prospective planning of temporal orders of items that are not physically visible. For example, prospective working memory has been documented in NHPs using paradigms that require planning ahead by masking future items of a sequence or require comparisons of the relative rank of items that appeared in different lists [19,31–35]. Our results extent these studies by suggesting that stronger working memory performance as measured with a delayed match-to-sample-task is linked to the mental ability for flexible re-indexing of objects to non-adjacent temporal positions.

### Swapping of sequences uses long-term memory

In addition to working memory, our results also suggest that pro-active mental re-indexing is more likely when the sequential structure of object relationships has been more firmly learned in long-term memory. Subjects more likely pro-actively swapped object B and D in context 2 the better they had learned the sequence in context 1. This result suggests that

pro-active swapping is a behavioral strategy that becomes available once a sequence is sufficiently well represented in memory [36]. A computational study supports this conclusion by quantifying the beneficial role of storing sequences in long-term memory compared to storing the same information in ongoing recurrent activity slots in working memory [26]. Using long-term memory avoids capacity limitation of short-term memory and thereby allows for more complex sequential structures. Our results suggest that the ability to form a longer-term memory of the sequential structure was central for the ability to pro-actively swap objects within that sequence. When subjects had a poorer long-term memory of the sequence, i.e., when they only poorly learned the context 1 sequence subjects they equally likely applied (erroneously) a serial inference strategy, i.e., choosing at the second ordinal position the adjacent item C instead of item D in context 2 (Figs 2C and 3G–3I). This finding extends previous studies that document monkeys represent [11,15–18], and memorize long term [12,31] serially ordered items by their relative ordinal rank. Our results are consistent with these prior findings by suggesting that representing serially ordered objects by chaining neighboring objects and knowing their relative rank in the sequence is a 'default strategy' by NHPs that is applied to a sequence when it is (1) either not learned sufficiently deep, or (2) when there is no task requirement to infer a more abstract temporal structure that would support the mental re-indexing operations of non-neighboring positions. According to this interpretation, NHPs are able to infer the sequential structure with sufficient experience of the temporal structure and do not have a 'hard cognitive limitation' of inferring temporal structures from item sequences beyond representing serial chains of item-item associations [5,23]. This conclusion is consistent with the re-indexing framework discussed above. It is well supported by neurophysiological evidence of neurons in the frontal cortex as well as in the medial temporal cortex. In these brain areas neuronal responses have been found that are tuned to the ordinal rank of items in multi-item sequences [22,37–39]. Prefrontal rank-selective neuronal responses predict the specific items that a subject uploads at each position even when that item is erroneously uploaded and leads to an unrewarded choice [37,40]. The neural coding of a rank position independently of the specific item that is encoded at that rank in principle supports a flexible assignment of items to different ranks, documenting that neuronal representations are not limited to serial item-item associations.

Our finding that rhesus monkeys are able to learn more than 6–8 *new* 5-object sequences every experimental session is an empirical finding that was not expected from the literature, which often trains shorter sequences, or trains the same sequence over multiple days. We speculate that the key determinant of the successful behavioral performance in our study – the ability of NHPs to flexibly re-index objects to non-adjacent positions in learned sequences – was facilitated by various design features of our task. Firstly, NHPs were exposed in each behavioral session to multiple sequence pairs with context 1 and the swapped items of context 2 already early during training. This design aspect ensured the swapping rule was not a rare exceptional task feature, but an integral part of their daily task environment, motivating them to figure out how to complete swapped sequences in context 2 in order to receive rewards. Secondly, our task paradigm enforced the rule 'retouch-the-last-correct-item after-an-error', which ensured that erroneously made serial object connections were not left unnoticed by were corrected immediately by the correct pair (see S1 Movie). Lastly, the task paradigm provided immediate performance feedback of every choice in the form of the halo feedback and the stepping forward or backward of the slider position of the progress bar (Fig 1A).While only the halo or only the slider progress would have been sufficient to provide feedback information this design aspect may have enhanced the likelihood subjects recognize when an object was chosen erroneously, which may facilitate learning.

## Conclusions

Taken together, our results show that NHPs are able to flexibly re-index objects to a non-adjacent position within 5-object sequences once they have been learned at high proficiency. This mental flexibility suggests that NHPs form non-spatial cognitive maps and use them to mentally manipulate items during goal-directed behavior [7]. We speculate that this mental capacity will have evolved in NHPs to support a higher level of adaptiveness of behavior during many serially organized behaviors beyond arranging visual objects in novel temporal relationships [2].

## Materials and methods

### Ethics statement

All animal and experimental procedures complied with the National Institutes of Health Guide for the Care and Use of Laboratory Animals and the Society for Neuroscience Guidelines and Policies and were approved with the approval number M1700198 by the institutional review board Vanderbilt University Institutional Animal Care and Use Committee (IACUC).

### Experimental procedures

Four male rhesus macaques (Monkey S: 10 yrs/12.6 kg; Monkey B: 10 yrs/10.8 kg; Monkey K: 12 yrs/11.9 kg; Monkey J: 13 yrs/ 12.9 kg) were used in this study. They performed the experimental task in their housing cage using cage-mounted touch screen stations [27] (Fig 1A). Visual display, behavioral response registration, and reward delivery were controlled by the Multi-Task Suite for Experiments (M-USE) [41]. M-USE is an open-sourced video-engine based Unity3D platform that is integrated with a touchscreen, a video camera system and reward delivery hardware.

The task required learning the sequential temporal order of sets of five objects. We generated novel sets of objects for every new sequence by randomly assigning each object different features from up to 10 different feature dimensions using multi-dimensional 3D rendered so-called Quaddle objects [42]. We used Quaddle 2.0 objects that vary in 10 feature dimensions (e.g., the shape, color, body pattern, different arm orientations, the presence of a head, etc.), each with >10 possible feature values (different body shapes, variable colors, etc.) (Fig 1C). The objects were generated using the software Blender and custom Python scripts that are freely available online [28]. For the experiment, object colors were chosen to be equidistant within the perceptually defined CIELAB color space. Objects were presented on an Elo 2094L 19.5″ LCD touchscreen with a refresh rate of 60 Hz and a resolution of 1,920 × 1,080 pixels, rendered at 2.9–4.2 × 2.4–4.7 cm on the screen.

For each sequence, a new contextual background image was displayed. Context images were generated with DALL·E 2, an AI system developed by OpenAI, using a text prompt for obtaining images from the categories rocks and sand. We applied color filters to images to obtain a larger number of distinct context backgrounds, ensuring each color filter maintained a fixed luminance value of 50. The colors were selected by evenly spacing them around the CIELAB color wheel, ensuring a diverse range of hues. The brightness of the background images was adjusted to be ≤50% of the HSL scale.

### Task paradigm

On each trial six objects were presented at a random location at equal eccentricity relative to the center of the screen (Fig 1A). Five of the objects were assigned a unique ordinal temporal position in the sequence, while a sixth object was a sequence-irrelevant distractor. Each sequence was presented for a maximum of 15 trials. In each trial subjects had a up to 15 choices to complete the sequence and earn fluid reward by choosing objects (by touching them with their hands, *see* S1 Movie) and receiving either positive feedback (a yellow halo and high pitch sound) for correct choices, or negative feedback (a transient grey halo and low pitch sound) for touching an object at an incorrect temporal position. After an erroneous choice, subjects had to re-choose the last correct object in the sequence before searching for the next object in the sequence (see S1 Movie). When a trial was completed, the objects were removed from the screen and a new trial was started with the objects displayed at random new locations equidistant from the center of the display. For each correctly chosen object, the slider position of the slider progress bar on top of the screen stepped forward. Successful completion of a sequence always completed the slider progress bar and resulted in a water reward. When a sequence was not completed no fluid reward was given and the slider progress bar was set back to 25% of the slider range for the subsequent trial with the same objects at new random locations. Changes of the slider bar were not instrumental for task performance. The two purposes of the slider bar were to provide additional visual feedback to the subjects about whether their object choice was correct or erroneous (the same feedback

was evident from the visual halo feedback), and how many choices they were away from completing the sequence and receiving reward. Subjects could have ignored the slider bar progress. However, previous studies suggest that secondary feedback in the form of symbolic assets such as visual token or the progress-status towards receiving primary reward, provide motivational incentives for subjects to stay on task and put effort in completing the current task [43]. Consistent with this behavioral evidence the gains and losses of symbolic assets (token or slider status changes) causes neuronal activity changes in the orbitofrontal cortex, striatum, and amygdala, where they serve motivational roles and modulate reinforcement learning [44].

After completing 15 trials on a sequence A-B-C-D-E we changed the background context and swapped the order of object B and D, requiring performing the sequence A-D-C-B-E for the next 15 trials (Fig 1C). We refer to the initial and the swapped sequence as a 'sequence pair'.

**Delayed presentation of novel and familiar sequences and same or different contexts.** Each session began by presenting four or five sequence pairs in a first set, followed by interleaving 120 trials of a delayed match to sample task, and followed by a second set with five or six sequence pairs (Fig 1B). In this second set one of the sequence pairs was novel, and four or five pairs were repeated sequence pairs from the early first set of sequences presented prior to the delayed match-to-sample task (Fig 1B). Fifty to 75% of sequences in the repeated set were presented with the same context, while the remaining sequences were randomly presented with different contexts.

**Distractor objects.** In each trial, one of six objects was a sequence-irrelevant distractor. Feature dimensions of the distractor were chosen to have a high degree of feature similarity to the second object of the initial sequence (which then became the fourth object in the swapped sequence). The sequence-irrelevant distractor object differed from the sequence relevant object at ordinal position two (object B) in only three features, while the other objects of the sequence did not share features (Fig 1C). Analysis results (S5 Fig) showed that subjects confused the distractor object most likely with object B in the first sequence in context 1 (second ordinal position) and also with object B in the second swapped sequence in context 2 (fourth ordinal position).

**Interleaved delayed-match-to-sample working memory task.** After the first and before the second set of sequences, subjects performed a delayed match-to-sample task (DMTS) for 120 trials (Fig 5A). The DMTS task is part of the M-USE platform [41]. DMTS used Quaddle 1.0 stimuli that were composed of features from four feature dimensions (nine different body shapes, 10 different arm styles, nine different body patterns, eight different color). In each block of the experiment, either two feature dimensions shared the same feature (high similarity), or none of the dimensions shared any features (low similarity). The DMTS trial presented a sample object for 0.5 s, followed by a delay of 0.5, 1.25 or 1.75 s, before two or three test objects were shown. One of the test objects matched the sample and when touched resulted in a yellow halo, a high pitch sound and a token reward (a green circle) that was added to a token bar [41]. Choosing non-matching objects resulted in a grey halo, a low-pitched tone, and a grey token that was subtracted from tokens available in the token bar. The token bar contained three placeholders for tokens and flashed white/red when all three tokens were completed, resulting in the delivery of fluid reward.

**Analysis of sequence learning.** In each trial subjects could make maximally 10 errors to reach the last, fifth object in a sequence or else a new trial started with the same objects in new positions. Up to 15 trials were shown with the same sequence. We quantified learning as the proportion of trials in which an object sequence was completed. A sigmoid fit was applied to the completion rates across trials from all subjects. The trial at which learning reached criterion performance was defined as the first trial at which subjects completed 80% of trials correctly. To assess overall performance, we calculated the average completion rate for each subject's sessions. We defined well performed sequences when more than 80% of the trials were completed; blocks falling below this threshold were considered poorly performed. To evaluate the consistency of learning at each ordinal position we calculated the proportion of correct choices for objects at each ordinal position across trials within a block. For each ordinal position, we identified the trial at which subjects achieved 80% correct object choices.

To analyze how learning over trials correlated with changes in choice reaction times, we calculated the average time of subsequent choices for each ordinal position (S2 Fig). We plotted the reaction times separately for trials before and after reaching the trials-to-80% criterion completion rate to investigate the effect of learning on reaction time. A Welch $t$ test was applied to compare the reaction time differences before and after the learning point at each ordinal position.

**Analysis of errors.** To evaluate how subjects learned, we quantified the decrease in the proportion of different error types across trials (S1D and S1E Fig): (1) An exploration error occurred when choosing an incorrect object among objects not yet learned; (2) A rule-breaking error involved incorrectly re-selecting or failing to re-select the last correctly chosen object after an error, or re-selecting a previously correctly chosen object; (3) A distractor error occurred when choosing the distractor object; (4) A perseverative error involved choosing the same incorrect object as in the last choice before reselection. We calculated the overall proportion of each error type relative to all errors as a function of the trial in a block. To evaluate how fast subjects reduced each type of error over trials we fitted an exponential decay function across 15 trials of the form $(a * \exp(-b * x) + c)$, where $a$ represents the amplitude of decay, $b$ represents the speed of decay (plotted in S1E Fig), and $c$ represents the asymptotic baseline. The same decay function was applied to each session individually, and the speed of decay was plotted to compare the rate at which different error types diminished. Statistical analysis was performed using one-sample $t$-tests to compare the decay factor to zero, and a Welch $t$ test was applied to compare each pair of error types.

**Analysis of re-ordered sequences with swapped object positions.** We tested how subjects learned the swapping of objects B and D from the first sequence A-B-C-D-E in context 1 to the second sequence A-D-C-B-E in context 2 (Fig 2). A sigmoid fit was applied to the completion rates over trials in context 1 and context 2. We calculated the average trials needed to reach learning criterion (≥80% completion rate) for each session across sequence pairs in context 1 and context 2, using a $t$ test to compare the differences. To investigate the effect of swapping on search time, we plotted the average reaction time for each ordinal position in both conditions (S2D Fig). Next, we analyzed which objects subjects chose after they incorrectly chose object B at the second ordinal position in context 2. A Welch's $t$ test was used to compare the transition probability of objects C, D, or E. We also calculated the probabilities of choice separately for well-performed and poorly performed blocks. To quantify how likely subjects chose objects at each ordinal position in context 1 and context 2 before they reached the 80% learning criterion, we calculated transition choice probabilities after choosing object A (Fig 2D and 2E) and after choosing correctly object D in context 2 (Fig 2F). We used $t$-tests to compare whether subjects inferred the correct swapped order in context 2 after choosing object D at the second ordinal position (choosing C), or whether they confused the sequential ordering in context 2 with the order from context 1 (and incorrectly chose E). To account for potential unequal variances in choice probability, we employed Welch's $t$ test. We used Bonferroni correction for correcting $p$-values for multiple comparisons or we used Benjamini–Hochberg FDR correction when this is explicitly stated.

**Analysis of individual re-ordering responses using Bayesian estimation.** We use a Bayesian framework to quantify the uncertainty surrounding each participant's distribution of choices of objects B, C, D, E, and the distractor object at the second ordinal position in context 2. This approach quantifies the full posterior distributions over the parameters of interest, rather than point estimates alone. The Bayesian framework also allows expressing and updating our prior beliefs about choice tendencies as data accumulate. We modeled each subject's aggregated counts of object choices across sessions as arising from a multinomial likelihood governed by a probability vector $\theta = (\theta_B, \theta_C, \theta_D, \theta_E, \theta_{distractor})$. A Beta prior distribution with initial values of $\alpha = 2$ and $\beta = 5$ was imposed on each component of $\theta$, limiting the choice probability to 0–1. The joint posterior for $\theta$ combined the multinomial likelihood distribution of observation $y$ in each subject $k$: $P(\mathbf{y} \mid \boldsymbol{\theta}) \propto \prod_k \theta_k^{y_k}$, with the product of Beta prior: $P(\boldsymbol{\theta}) \propto \prod_k \theta_k^{\alpha-1} (1 - \theta_k)^{\beta-1}$. Likelihood was estimated from the choice probability of each object at the second ordinal position (after object A was chosen) for all trials in context 2 that occurred before the learning criterion (80% completion rate) was reached. To approximate the posterior distribution, we implemented a Metropolis–Hastings Markov Chain Monte Carlo (MCMC) sampler. Each chain was initialized at the

maximum likelihood estimate and iterated for 200,000 draws of $\theta$, discarding the first 100,000 as burn-in. Proposals were generated via a Gaussian random walk with step size 0.05. To assess the convergence of our Monte-Carlo Markov Chain (MCMC) procedure, we examined both the running mean of the parameter estimates and the Gelman–Rubin diagnostic (S6 Fig). The running mean of theta parameters across iterations initially fluctuated but stabilized after approximately 100,000 iterations for each of the subjects (S6A Fig). We computed the Gelman–Rubin convergence diagnostic ($\hat{R}$) from multiple parallel chains initiated at dispersed starting points. The resulting values closely approached 1.0 for all key parameters ($\hat{R} < 1.05$), at a level that was well below a threshold of 1.1 that is considered satisfactory convergence (S6B Fig) [45–48].

**Analysis of the effect of learning proficiency and task experience on swapping behavior.** We quantified how the proficiency of performing a sequence (the average completion rate in context 1) affected the likelihood of pro-active swapping behavior in context 2. We used the Welch $t$-tests with Benjamini–Hochberg false discovery rate (FDR) correction to compare the values indexing the likelihood of showing incorrectly a serial inference strategy or a pro-active swapping strategy in context 2 for sequences performed at low versus high completion rates in context 1. We also correlated these values on a sequence-by-sequence basis using linear regression and used linear regression of the performance across sessions to discern the influence of experience over the course of the experiment.

**Analysis of effect of memory on swapping.** To quantify the effect of memory on sequence learning we tested whether sequences that were repeated late in the session were performed better than novel sequences performed late in the session (Fig 5). Using paired $t$-tests, we compared how fast sequences were performed at learning criterion (≥80% completion) that were shown early ('new early'), or repeated late in the session ('repeated'). We also included novel sequence pairs that were presented late in the session ('new late'), or introduced newly late in the session ('new late'). Next, we analyzed how memory influenced pro-active swapping and retro-active swapping behavior. We calculated two measures: a value indexing the likelihood that subjects chose object C instead of object D in context 2 at the second ordinal position (an incorrect serial inference), defined as: $p(choosing\ D - choosing\ C\ |\ erroneously\ choose\ B)$; and we calculated a value indexing the likelihood of choosing object D at the second ordinal position in context 2, reflecting pro-active swapping: $p(choosing\ D - choosing\ B\ |\ correctly\ choose\ A)$. $T$-tests were used to quantify the difference of these strategies in these repeated versus new/late new sequence pairs.

**Analysis of the relation of working memory and sequence learning.** The delayed match-to-sample task presented a sample stimulus, followed by a blank delay lasting 0.5, 1.25, or 1.75 s, followed by the presentation of three objects (2 distractors and 1 target object matching the sample stimulus) (Fig 6). The distractor stimuli were either similar or dissimilar to the target stimulus, because they shared, or did not share, features with the target object. We analyzed the overall accuracy for low/high similarity and across delays. Linear regression analysis showed no decline across delays so that we averaged the accuracy across conditions for individual sessions and each monkey (S4 Fig). To analyze the relationship of working memory, sequence learning and the ability to anticipate the swapped object order in context 2, we applied individual linear regression, and a linear mixed model fit to the performance across sessions.

**Analysis of context repetition on sequence learning.** In a subset of sequence pairs shown late in the session, we repeated the object sequences from early in the session but on a different background context. To test how the difference of contexts versus the similarity of contexts affected performance and the likelihood to observe pro-active swapping behavior in context 2 we determined the difference in accuracy between early and late (repeated) sequences with the same and different contexts (Fig 6D–6F). Paired $t$-tests were used to compare the accuracy between conditions.

**Analysis of choices of the distractor.** For every 5-object sequence the task presented a sixth distracting object that was irrelevant to the sequences and shared three features with the object B. We tested whether the distractor object was more likely chosen at the ordinal position at which it shared features with object B of the sequence than at other ordinal positions. We tested for differences in the proportion of distractor choices in context 1 and 2 using a proportion $Z$-test (S5 Fig).

 

## Supporting information

**S1 Fig. Overall learning performance.** (**A**) Prop. correct choices for each ordinal position (*y*-axis) and trial (*x*-axis) for each of the four subjects for novel sequences in context 1. (**B**) Probability to reach 80% correct choices (*y-axis*) for each ordinal position (diff. colors) across trials (symbols mark different subjects). (**C**) Avg trial position that subjects completed in >80% of sequences (*y*-axis) for each ordinal position (diff. colors). Error bar indicates mean and 95% CI. Ordinal position 1: 1.03 trials (±0.03), 2: 1.27 (±0.09), 3: 1.85 (±0.19), 4: 2.62 (±0.32), and 5: 3.66 (±0.43). (**D**) Exploration errors, distractor errors (choosing the distractor object that was shown as a sequence-irrelevant sixth object in the display), and rule-breaking errors decreased over trials. Data were fit with an exponential decay function ($y = a * \exp(-bx) + c$). (**E**) Mean (and 95% CI) of the decay factor (b) for each error type: Exploration error (0.14 [0.12, 0.17]), Rule breaking error (0.0041 [0.0025, 0.0058]), Distractor error (0.15 [0.12, 0.19]), and Perseverative error (0.022 [0.014, 0.029]). Exploration and distractor errors had a similar decay rate ($p = 0.66$). Stars denote significance levels for pairwise comparisons (Welch's *t*-tests). Decay factors for all error types were significantly greater than zero, confirming a decrease in all error types over the course of learning (t test against zero, *p*-values: Exploration error: $6.6 \times 10^{-18}$; Rule breaking error: $5.4 \times 10^{-6}$; Distractor error: $7.3 \times 10^{-16}$; Perseverative error: $4.3 \times 10^{-8}$). The data underlying this figure can be found in the S1 Data file.
(DOCX)

**S2 Fig. Accuracy and reaction times to objects across ordinal positions in context 1 and 2.** (**A**) Subjects reached 80% completion rate earlier in context 2 (yellow). (**B**) Trials to reach 80% completion (learning speed) in context 1 and 2. (**C**) Average reaction times for correct touches before and after learning point. Before learning points, ordinal position 1–5, respectively (Mean±95%CI): 0.84±0.06, 1.14±0.06, 0.57±0.04, 0.59±0.04, 0.62±0.04; After learning points: 0.66±0.04, 0.76±0.05, 0.72±0.05, 0.94±0.06, 0.86±0.05. Welch's *t* test was applied to each ordinal position, and significant difference was found at the first and the last ordinal positions (Position 1: $p < 0.00001$, Position 2: $p = 0.5520$, Position 3: $p = 0.2550$, Position 4: $p = 0.3416$, Position 5: $p = 0.0384$). (**D**) Average reaction times for correct touches before (blue) and after swap (yellow) for each subject. No significant difference was found in all ordinal positions. The data underlying this figure can be found in the S1 Data file.
(DOCX)

**S3 Fig. Memory of sequences improved performance.** (**A**) Sequences (here: context 1 sequences only) repeated after the working memory task are performed better. In the new early condition, the 80% completion rate was reached after 3.93±0.50 trials (Mean±95%CI), whereas, in the repeat condition, it was significantly quicker at just 2.13±0.32 trials (New early versus Repeat: *p*: $1.19 \times 10^{-8}$). For new context 1 sequences introduced late (to control for 'time in the session'), the avg. trial to reach the 80% criterion was 2.87±0.38, demonstrating that while there was some improvement in performance for new sequences in the later set, the most substantial gain was observed for the repeated sequences (New early versus New late: *p*: 0.0014; Repeat versus New late: *p*: 0.0052). (**B**) Choice reaction times for correct choices at each ordinal position were calculated to check if there were any difference in three conditions (New early: 0.79±0.04; Repeat: 0.81±0.04; New late: 0.77±0.031). We found a slightly slower reaction time for the repeat blocks when compared with New late blocks (*p*: 0.039), but no other comparisons were significant (New early versus Repeat: *p*: 0.33; New early versus New late: *p*: 0.35). The data underlying this figure can be found in the S1 Data file.
(DOCX)

**S4 Fig. Working memory (delayed match-to-sample) performance.** (**A**) Task performance across varying object feature similarities and delay times. Objects shared features in either two feature dimensions (high similarity), or did not share features of any dimension (low similarity). Chance performance was set at 0.33. Accuracy was significantly higher for the Low Similarity condition compared to the High Similarity condition. High Similarity condition: Intercept: 0.4606; Slope:

0.0097; *p* = 0.6670; Low Similarity condition: Intercept: 0.7172; Slope: −0.0175; *p* = 0.3772. The overall average accuracy across all conditions was 0.59 ± 0.16 (Mean ± 95% CI), notably above chance (0.33). For the 0.50 s delay condition, performance was 0.47 ± 0.12 (High Similarity) and 0.71 ± 0.22 (Low Similarity); for the 1.25 s delay condition, it was 0.46 ± 0.12 (High Similarity) and 0.70 ± 0.19 (Low Similarity); for the 1.75 s delay condition, 0.48 ± 0.14 (High Similarity) and 0.68 ± 0.18 (Low Similarity). **(B)** Individual subject performance across all similarity and delay time conditions. Subject-level data remained consistent, with similar patterns observed between high and low similarity conditions, regardless of delay time. The data underlying this figure can be found in the S1 Data file.
(DOCX)

**S5 Fig. Learning to ignore distractor object. (A)** Proportion of distractor choices across trials in context 1 (*left* panel) and context 2 (*right* panel). The mean value is shown as the rightmost data point (Mean ± 95% CI, Before swap: 0.090 ± 0.010; After swap: 0.061 ± 0.006). **(B)** Same format as **A** for distractor choices in sequences early (left) and late (right) in the session. The average is shown as rightmost data point (Mean ± 95% CI, Initial encounter: 0.076 ± 0.008; Repetition: 0.046 ± 0.004). **(C)** Distractor choices at each ordinal position in context 1 (distractor is similar to B at the second ordinal position) and in context 2 (distractor is similar to B at the fourth ordinal position). Two-proportion *Z*-tests were applied at each ordinal position for comparing the difference. Stars denote sign. level. (Ordinal Position: Before and After Swap, Mean ± 95% CI) 1: 0.13 ± 0.01; 1: 0.08 ± 0.01; 2: 0.29 ± 0.01; 2: 0.20 ± 0.01; 3: 0.19 ± 0.01; 3: 0.22 ± 0.01; 4: 0.22 ± 0.01; 4: 0.34 ± 0.01; 5: 0.17 ± 0.01; 5: 0.16 ± 0.01. The data underlying this figure can be found in the S1 Data file.
(DOCX)

**S6 Fig. Convergence diagnostics of Metropolis–Hastings Markov Chain Monte Carlo (MCMC) sampling. (A)** Running means of posterior theta over iterations for each subject and each object. All values stable after burn-in. **(B)** Gelman–Rubin diagnostics for each subject and each object. All values are less than 1.1 representing good convergence. The data underlying this figure can be found in the S1 Data file.
(DOCX)

**S1 Movie. Example performance of the context-dependent object sequence learning task.** Nonhuman primates (rhesus macaques) performing trials of the 5-object sequence learning task and adjusting to the swapped object order when the context changes from context 1 to context 2. The left side of the display illustrates the correct sequential order (left top) and the display visible to the experimenter that mirrors the main events of the screen (left bottom). The visual displays and task is controlled by the M-USE software platform [41]. The display on the right shows a back view from inside the animal cage while the subject engages with a touchscreen station. The touchscreen station is mounted to one apartment cage compartment. The subject has free access to the Kiosk touchscreen station (described in [27]) for approximately 90–120 min each week-day to engage with the task. Subjects hold their hand and mouth close to a stainless-steel sipper tube that delivers fluid rewards once a 5-object sequence is completed. The slider position of the progress bar on top of the screen steps forward with each correct choice and backwards after errors. The movie has two parts. The first part shows example performance of subject 'NHP-K' of a sequence after the subject reached the learning criterion (completing sequences in >80% of trials). Then, performance is shown for a swapped sequence in context 2 (a bluish context 1 background switches to a green context 2 background). The second part of the movie (starting at approximately 1:08 min) shows the example performance of subject 'NHP-J' for a context 1 and then a context 2 sequence. The data underlying this figure can be found in the S1 Data file.
(MP4)

**S1 Data. Source data file with data underlying each figure panel.**
(XLSX)

## Author contributions

**Conceptualization:** Xuan Wen, Thilo Womelsdorf.

**Data curation:** Adam Neumann, Thilo Womelsdorf.

**Formal analysis:** Xuan Wen.

**Funding acquisition:** Thilo Womelsdorf.

**Investigation:** Xuan Wen, Adam Neumann.

**Methodology:** Xuan Wen, Adam Neumann, Thilo Womelsdorf.

**Software:** Xuan Wen, Seema Dhungana.

**Supervision:** Thilo Womelsdorf.

**Validation:** Xuan Wen, Thilo Womelsdorf.

**Visualization:** Adam Neumann, Xuan Wen, Thilo Womelsdorf.

**Writing – original draft:** Thilo Womelsdorf.

**Writing – review & editing:** Xuan Wen, Thilo Womelsdorf.

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
