## [Editor Report · Decision Letter 0]

Dear Dr Womelsdorf,

Thank you for submitting your manuscript entitled "Flexible Learning and Re-ordering of Context-dependent Object Sequences in Nonhuman Primates" for consideration as a *Short Report* by PLOS Biology.

Your manuscript has now been evaluated by the PLOS Biology editorial staff as well as by an academic editor with relevant expertise and I am writing to let you know that we would like to send your submission out for external peer review.

Once your full submission is complete, your paper will undergo a series of checks in preparation for peer review. After your manuscript has passed the checks it will be sent out for review. To provide the metadata for your submission, please Login to Editorial Manager (https://www.editorialmanager.com/pbiology) within two working days, i.e. by Feb 15 2025 11:59PM.

Kind regards,

Christian

Christian Schnell, PhD

Senior Editor

PLOS Biology

cschnell@plos.org

---

## [Decision Letter · Decision Letter 1]

Dear Dr Womelsdorf,

Thank you for your patience while your manuscript "Flexible Learning and Re-ordering of Context-dependent Object Sequences in Nonhuman Primates" was peer-reviewed at PLOS Biology. It has now been evaluated by the PLOS Biology editors, an Academic Editor with relevant expertise, and by several independent reviewers.

In light of the reviews, which you will find at the end of this email, we would like to invite you to revise the work to thoroughly address the reviewers' reports.

As you will see below, the reviewers think that the study is very well executed and provides important insights. However, Reviewer 1 is not convinced that the results are sufficiently novel for PLOS Biology. The concerns from Reviewer 2 and Reviewer 3 mostly relate to issues with the presentation, statistical analyses and the lack of details and clarity in the writing. Reviewer 2 also mentions that the small sample size is a concern for the generalization. We are asking you to address all reviewer concerns and in particular make a strong case how your findings extend beyond what some of the more neurophysiological studies in rodents and monkeys have shown previously to convince the reviewers about the advance and distinction with respect to those prior studies.

Given the extent of revision needed, we cannot make a decision about publication until we have seen the revised manuscript and your response to the reviewers' comments. Your revised manuscript is likely to be sent for further evaluation by all or a subset of the reviewers.

**IMPORTANT - SUBMITTING YOUR REVISION**

*Re-submission Checklist*

*Published Peer Review*

*PLOS Data Policy*

*Blot and Gel Data Policy*

Sincerely,

Christian

Christian Schnell, PhD

Senior Editor

PLOS Biology

cschnell@plos.org

REVIEWS:

Reviewer #1: These experiments show that macaque monkeys are capable of learning abstract slots or positions in a sequence. The evidence is ability to obey a general rule "swap slot 2 with slot 4". On the whole, I find the conclusion to be well supported, and some aspects of the results (e.g. specific link between working memory and rule following) are quite intriguing. On the whole, though, I feel that abstract serial position slots in animals are already well supported by physiological data for the monkey (papers from Wang group cited here) and mouse (e.g. Behrens group). I am not sure that enough is added by these behavioral data to justify publication in this journal.

I also have two relatively major (though addressable) concerns. First, I don't think a comparison of learning rates in contexts 1 and 2 (e. g. line 169 onward) is very telling - animals have already learned several things in context 1, so it is really not surprising that there is some generalisation to context 2. Similarly, I don't think it is very useful to show that correct swapping is more common when a sequence was previously learned, earlier in the session (section beginning line 254) - that earlier experience was for both contexts 1 and 2, so of course, context 2 (with swapping) also gains from that earlier experience.

Minor comments:

Line 119. I think "reset" is not a good word to use - presumably the counter does not go back to the start, as the animal is supposed to repeat just one previous choice?

Line 133. This is not what is shown in Figure 1C.

Lines 149, Figure 1, Figure S1. A variety of apparently different 80% rules are used - it is confusing, and perhaps needs more careful explanation of each.

Figure 1, Line 262. It should be explicit that these data refer to context 1. The reference to Figure 1 in line 133 is confusing since Figure 1 presents context 1 results.

Line 589. Closing parenthesis missing.

Line 740. Figure 1E does not seem to show mean trials to reach completion.

Figure 4E. What are colours and dashed black line?

Figure S1. Distractor error not defined here.

Figure S3. Should specify this is for context 1.

Figure S4. Similarity not defined here.

Reviewer #2 (Maël LEROUX): Review PloS Biology

This study examines whether rhesus macaques can flexibly reorder learned object sequences based on contextual cues. While primates can swap adjacent items in sequences, it remains unclear if they can reassign objects to non-adjacent positions. Using a touchscreen task, macaques learned to order five objects and adjusted their responses when a background change signaled a new sequence. They successfully re-indexed objects, particularly when the original sequence was well-learned, and performance correlated with working memory. This study presents an interesting and valuable contribution to our understanding of cognitive flexibility in nonhuman primates. The authors provide novel insights into how primates infer and manipulate sequential structures. The findings challenge traditional views on sequence learning, suggesting that monkeys can go beyond simple associative chaining to infer latent serial structures. These results have important implications for research on primate cognition, comparative psychology, and the evolutionary roots of structured thought.

However, there are several issues that I believe, once dealt with, could improve the manuscript. One issue is the current organization of the paper, where the methods appear at the end due to journal formatting. As a result, the experimental design is difficult to fully grasp until late in the manuscript, making it challenging for the reader to understand the logic behind the study. If restructuring is not possible, the authors should provide a more explicit description earlier in the manuscript, clearly explaining how sessions, sequences, and trials are structured - as they do L472-489. A more transparent outline of the learning process would significantly improve readability.

Additionally, the writing could be more fluid and accessible. While the study deals with complex cognitive concepts, the occasional use of jargon makes certain sections dense. The authors could simplify some descriptions, use more commas to clarify sentence structure, and ensure that key points are conveyed as clearly as possible.

To give an example, L105-109 could be clarified by (i) structuring the sentence as follow: "In computational models -COMMA- the latent temporal structure of experienced environments can be inferred and represented as a non-spatial cognitive map of item locations -FULL STOP- Such representation enables the flexible re-indexing of objects to different positions in this cognitive map (Behrens et al., 2018; Whittington et al., 2022)" and (ii) expliciting the terminology used. A more reader-friendly presentation would make the study's significance and findings more approachable to a broader audience, including researchers outside the field of comparative cognition.

The discussion on the neural basis of the cognitive operations described in the study is somewhat unclear. While drawing connections to neural mechanisms is valuable, it is not entirely evident how the behavioral findings directly map onto specific neural processes. The authors should clarify this link or consider whether it is necessary for the core argument of the paper.

Another concern is the small sample size. The study reports data from only four rhesus macaques, which limits the generalizability of the findings. It would be useful for the authors to clarify whether additional subjects were tested and, if so, why they were excluded. If only four monkeys were tested from the start, acknowledging this limitation more explicitly and discussing its implications for interpretation would be beneficial.

The statistical analyses also require more transparency. The authors primarily use Welch's t-tests, but this method is not ideal for such a small sample size, particularly for multiple pairwise comparisons.

At times, the manuscript simply states that a "t-test" was conducted without specifying whether it was Welch's or Student's (e.g. L559, 562), leading to some ambiguity. There are also inconsistencies in the choice of statistical tests—while paired t-tests are used in some comparisons (L582), they are not applied consistently. A more detailed explanation of why specific tests were chosen, along with a justification, is necessary. Additionally, the manuscript would benefit from a clearer description of the statistical tools used, including the R functions and packages. Providing the R script would further enhance transparency and reproducibility. Given the small sample, Bayesian analyses might be a more robust alternative and should be considered.

Overall, this study presents interesting findings and contributes to our understanding of cognitive flexibility in primates. Addressing these concerns would significantly strengthen the paper and improve its accessibility to a broader scientific audience.

Review by Maël Leroux, University of Rennes

In an effort to promote transparency and kindness in academic discourse, I have chosen to systematically sign my reviews. While I recognize that this may not be suitable for everyone, particularly early-career researchers, I believe that fostering respectful and constructive feedback is essential for advancing science and maintaining integrity within our community.

Reviewer #3: This paper presents a detailed experiment to investigate whether nonhuman primates can learn to re-order non-adjacent elements in an order element. I am not an expert on core cognition, so I will not focus my review on the experimental design - although it sounds robust, thoroughly designed and relevant to me-, but rather on the manuscript itself.

I think that given PlosBiology is not a specialized journal and has a large audience, the manuscript would benefit from more pedagogical writing. Indeed, at the moment, I think that it is written more for an audience of experts, which makes it hard to follow in some places. In particular, the relevance of the study to science is not clear. I would appreciate a few sentences at the beginning of the intro and at the end of the discussion to frame why this was important to study. This does not need to be long, but 1-2 sentences with clear examples could be of good help to evaluate the impact of the paper. I also struggled to follow every step of the study, since some information about the design is missing. Given that this is a dense paper with lots of different analyses and results, I think it would be good to "take the reader by the hand" a bit more and better explicit what was done, and why.

I also have questions about the statistics - I am not asking the authors to redo them, but to justify their choice - why did you use mostly non-parametric tests (Z-test, t-tests) and not models, where you could have controlled for the identity of the subject or the sequence? You also did not mention how you processed the experimental data (e.g., which software did you use?) and how you conducted the statistical analyses (R? If so, what packages?).

Below you'll find more specific comments:

L73 "Flexible mental operations": Maybe keep the same terminology to avoid confusion in the reader

L74-75: Maybe give an example in humans? It is not clear how these are necessarily ordered behaviours, and how flexible mental operations are crucial

L77 "latent temporal structure": I would detail what this is exactly, especially since it is a crucial concept in your study (maybe give an example?)

L86: ordering of items?

L91: what is a "reverse play 3-item sequence"?

L92-94: It is not clear how the studies you mention point to these limitations

L96: You say "This re-indexing can be achieved by swapping the relative rank of adjacent items" but in your example, A and C are not adjacent, right?

L113-115: This sentence is a bit too long and complex

L124-126: I am not sure, at this stage, what a session is. Does it mean that each individual had to learn a new pair for each new work session?

L131 "by erroneously choosing the object C that is adjacent to object B in context 1": I am not sure what this mean, can you be more specific?

L134 "serial inference strategy": What is this?

L141: In line with the previous comment, I would appreciate a bit more details about what a session is. Does it mean that monkeys had to learn several pairs? Or just one pair per ind, across several sessions?

L148-149: I don't understand this criteria. You are looking, in a sequence, for the first trial for which they have at least 80% of good answers? Or do you take 80% of all the trials with less than 10 errors, but then what do you do with them? Please clarify. I also think that 10 errors out of 15 choices is a lot and does not spontaneously look like success... Why this threshold?

L153 "80% completion rate": Is this the same as the criteria above? Here, it sounds as if you look at the first trial for which you have 80% of correct choices, but that contradicts the "10 or less error" criteria, as 10 errors / 15 choices is less than 80% success. Also, I don't understand the link with the "less than 5 errors per trial", this is not 80% success. Thank you to clarify these

L181-182: This part is not clear to me. It sounds as if some subjects spontaneously tried B and then saw it did not work so they tried C because it is the closer, vs some subjects immediately tried D. That's probably not what happened, and it is not clear why they would choose D over E, for example. Please clarify this sentence.

L184-186: I would also like to know, after they understood that D came second, what they did with B. Did they immediately assume that it was replacing D?

L188-190: I don't understand how they assumed this immediately. Was there a previous training before? Why would they go for D directly, instead of C and E for example? If this is not an immediate answer, then clarify this by stating after how many trials they adapt each strategy

L205-212: Over how many trials? Are you looking at all the trials, just the first ones? This needs to be clarified

L212-215: Shouldn't this go in the previous paragraph, where you detail the strategies?

L231 "Across all sequences": At this stage, I still don't understand if the sequences are the same series of objects for each individual, or if the objects and order change between each series. Because if the objects and order do not change, then one can simply argue that they have learned the orders of the objects for each context and just re-do them right after several sequences

L252: This info is very important and needs to come before. Also, you need to clarify your results section so that we understand over which dataset you perform each analysis

L258: What is the point of the match-to-sample task here?

L261-262: What does it mean? You did not introduce what a "repeated sequence" is

L295-296: I still don't understand what this does here. What do you match to what, for what?

L297 "WM": Introduce this abbreviation before using it.

L316-317 "the swapping occurred... error": But only after a few errors, no? They did not spontaneously change to the D without previous exposure to this task?

L322-323 "for more sessions later in the experiment": This is confusing, it sounds as if something happening after influenced what happened before. Reformulate

L414 "stepping forward or reset of the slider position of the progress bar": Are you sure that monkeys actually use these cues? Please provide a reference.

L483-484: At this stage, it is still not clear what the difference between a sequence and a trial is

L492: what is a sequence pair, is it context 1 + context 2?

L496-497: Why did the context vary in one condition but not in the other?

L504-505: If they didn't pay attention to the features of the objects and still managed to solve the task, what would be the bias or the problem in the interpretation? Or in other words, why is that an important control?

L509-519: It is still not clear what the point of this is. It is to ensure a time gap between the early and late sequences?

L523: It is only now that I understand that a sequence is a set of objects in a specific order, and not a set of trials. Is that right? This needs to be explicited somewhere as it relies too much on the understanding of the reader and some parts of the text makes it confusing

L548 formula: What are a, b and c here?

Figures: make sure that they are adapted to the vision of colour-blind people.

Fig 1C : in context 1, 2 or both? It sounds from the legend of Fig1 as if this is only in context 1. How do you explain that they learn A very easily and then it dramatically drops for B?

Fig 1D: "for at least the next 3 trials" is this a new criteria? Please unify across the manuscript.

Fig 2F and 2G: it is not clear whether the "first choices in a trial prior..." graph belongs to F or G, maybe redesign this figure or give another letter to this graph to make it clearer.

Fig 3J: this graph is hard to process, can you explicit it more, directly on the figure or/and in the legend?

Fig 4: Can you make it clearer, on the y-axes, in which direction the preferences are going? For example, in Fig4B, "D vs B": is the preference for D below or above zero?.

---

## [Decision Letter · Decision Letter 2]

Dear Dr Womelsdorf,

Thank you for your patience while we considered your revised manuscript "Flexible Learning and Re-ordering of Context-dependent Object Sequences in Nonhuman Primates" for publication as a Research Article at PLOS Biology. This revised version of your manuscript has been evaluated by the PLOS Biology editors, the Academic Editor and the original reviewers.

Based on the reviews and on our Academic Editor's assessment of your revision, we are likely to accept this manuscript for publication, provided you satisfactorily address the following data and other policy-related requests:

* We would like to suggest a different title to improve its accessibility for our broad audience:

Non-human primates can learn serial sequences and reorder context-dependent object sequences

* Please add the links to the funding agencies in the Financial Disclosure statement in the manuscript details.

* DATA POLICY:

Regardless of the method selected, please ensure that you provide the individual numerical values that underlie the summary data displayed in the following figure panels as they are essential for readers to assess your analysis and to reproduce it: 2BDEF, 5BCDF, 6DEF, S1CE, S2 and S3.

* CODE POLICY

We expect to receive your revised manuscript within two weeks.

*Published Peer Review History*

*Press*

Sincerely,

Christian

Christian Schnell, PhD

Senior Editor

cschnell@plos.org

PLOS Biology

Reviewer remarks:

Reviewer #1: The authors have addressed my concerns, and though I expressed reservations over the novelty of the conclusions, the support from other reviewers (along with the authors' rebuttal) suggests that the paper is suitable for PLOS Biology.

Reviewer #2 (Maël Leroux): I would like to sincerely commend the authors for the exceptional effort they have put into revising their manuscript. The extent and depth of the revisions are remarkable. The manuscript has been thoroughly transformed, both in terms of structure and clarity, and it is now significantly more readable, accessible, and compelling.

The authors have responded to the previous comments with great care and professionalism. The additional clarity around the structure of sessions, sequences, and trials greatly improves the reader's ability to follow the logic of the experimental design. The writing is now more fluid, with reduced jargon and clearer sentence construction, making the core ideas more accessible to a broader audience. The authors have also addressed concerns about the neural parallels with greater precision, and have clarified their relevance and scope within the context of the paper.

Furthermore, the statistical transparency has been substantially improved. The clarifications regarding sample size, choice of statistical tests, and the inclusion of more detailed methodological and analytical descriptions—including code references—are highly appreciated. The additional statistical analysis, although not expected, really improve the manuscript and make the authors' findings much more convincing for researchers outside their field. These changes greatly strengthen the manuscript's reproducibility.

In short, the authors have gone above and beyond in addressing the reviews, resulting in a much stronger and clearer manuscript. I now fully support the publication of this work in its current form and recommend acceptance without further revision.

Review by Maël Leroux, University of Rennes

In an effort to promote transparency and kindness in academic discourse, I have chosen to

systematically sign my reviews. While I recognize that this may not be suitable for everyone,

particularly early-career researchers, I believe that fostering respectful and constructive feedback

is essential for advancing science and maintaining integrity within our community.

---

## [Editor Report · Decision Letter 3]

Dear Dr Womelsdorf,

Thank you for the submission of your revised Research Article "Non-human primates can flexibly learn serial sequences and reorder context-dependent object sequences" for publication in PLOS Biology. On behalf of my colleagues and the Academic Editor, Simon Townsend, I am pleased to say that we can in principle accept your manuscript for publication, provided you address any remaining formatting and reporting issues. These will be detailed in an email you should receive within 2-3 business days from our colleagues in the journal operations team; no action is required from you until then. Please note that we will not be able to formally accept your manuscript and schedule it for publication until you have completed any requested changes.

When you attend to those requests to come, please also make sure to provide a DOI to the github repository and modify the Data Availability Statement accordingly.

PRESS

Sincerely, 

Christian

Christian Schnell, PhD

Senior Editor

PLOS Biology

cschnell@plos.org